# Simulation-based inference of developmental EEG maturation with the spectral graph model
Danilo Bernardo [1] ✉, Xihe Xie[2], Parul Verma[3], Jonathan Kim[1], Virginia Liu[1], Adam L. Numis[1], Ye Wu[4], Hannah C. Glass[1,5,6], Pew-Thian Yap [4], Srikantan S. Nagarajan[3] & Ashish Raj[3]

The spectral content of macroscopic neural activity evolves throughout development, yet how this maturation relates to underlying brain network formation and dynamics remains unknown. Here, we assess the developmental maturation of electroencephalogram spectra via Bayesian model inversion of the spectral graph model, a parsimonious whole-brain model of spatiospectral neural activity derived from linearized neural field models coupled by the structural connectome. Simulation-based inference was used to estimate age-varying spectral graph model parameter posterior distributions from electroencephalogram spectra spanning the developmental period. This model-fitting approach accurately captures observed developmental electroencephalogram spectral maturation via a neurobiologically consistent progression of key neural parameters: long-range coupling, axonal conduction speed, and excitatory:inhibitory balance. These results suggest that the spectral maturation of macroscopic neural activity observed during typical development is supported by age-dependent functional adaptations in localized neural dynamics and their long-range coupling across the macroscopic structural network.

The human brain undergoes dynamic and complex maturational processes throughout childhood, which starts prenatally[1] and remains highly dynamic, especially from infancy to early childhood[2]. Morphological transformations of the brain across the lifespan are well established[3] and underpin well-described structural and functional network modifications occurring at various temporal and spatial scales during development[4]. However, the mechanisms linking morphological and structure-function remodeling to electrophysiological developmental changes are unknown. This gap is critical, as deviations from the typical electrophysiological maturation are associated with a spectrum of developmental disorders, including autism and epilepsy[5–7].

Since Berger first reported on differences between the electroencephalogram (EEG) of children and adults, the emergence of canonical brain oscillations such as the posterior dominant rhythm (PDR) and their expected progression has been well documented[8–17]. Recently, parameterization of aperiodic EEG spectral components has further characterized age-related electrophysiological changes[18–22]. Despite the critical importance of developmental electrophysiological

activity in investigating typical brain development and in clinically monitoring neurodevelopmental conditions[23–25], the mechanisms underlying the spectral maturation of macroscopic brain activity remain unclear.

In parallel with electrophysiological developmental studies, developmental neuroimaging studies have yielded insights into brain network development from in utero through adulthood[26,27], and have demonstrated a tight coupling between structural and functional networks across development[28–33]. Thus follows the intuition that maturational changes of structural and functional connectivity underlie changes in the developmental spectral content of brain oscillations[13,24,34–36]; however, a mechanistic understanding of this relationship has been elusive. In this regard, whole-brain simulations involving modeling of macroscopic neural activity across varying spatiotemporal scales have demonstrated promise in elucidating mechanisms of structure-function coupling[37], yet, to the best of our knowledge, no prior studies have aimed to model the brain-wide electrophysiological spectral maturation seen in development. As EEG maturation reflects the refinement of structural and

[1]Department of Neurology and Weill Institute for Neurosciences, University of California, San Francisco, San Francisco, CA, USA. [2]Department of Neuroscience, Weill Cornell Medicine, New York, NY, USA. [3]Department of Radiology and Biomedical Imaging, University of California, San Francisco, San Francisco, CA, USA. [4]Department of Radiology and Biomedical Research Imaging Center, University of North Carolina, Chapel Hill, NC, USA. [5]Department of Pediatrics, University of California, San Francisco, San Francisco, CA, USA. [6]Department of Epidemiology and Biostatistics, University of California, San Francisco, San Francisco, CA, USA. ✉e-mail: dbernardoj@gmail.com

functional brain networks[13,36], spectral graph theory provides a principled foundation for modeling network-driven EEG spectral changes during development. Thus, we were motivated to apply the spectral graph model (SGM), which accounts for spatial propagation of local neural activity across the spectral graph of the structural connectome[38]. The SGM predicts spatiospectral activity in a parsimonious manner with seven biologically interpretable parameters (Table 1), thus making it well-suited for modeling the mechanistic principles of electrophysiologic maturation.

In this paper, we demonstrate the Bayesian inference of SGM parameters from a developmental EEG database to evaluate the population-based temporal evolution of SGM parameter space during the developmental period (Fig. 1). We leverage recent deep learning advances in simulation-based inference (SBI) that enable efficient estimation of the posterior distribution over SGM parameters[39], and demonstrate that age-associated changes in EEG spectra are described by a neurobiologically consistent, temporal progression of long-range coupling strength ($\alpha$), axonal conduction speed ($S$), and excitatory:inhibitory balance. Furthermore, we validate our approach by demonstrating age prediction from EEG spectra with a regression model incorporating these parameters.

## Results

### Subjects

To evaluate the developmental spectral maturation of macroscopic brain activity, we constructed a developmental EEG database containing EEGs from subjects ranging from 1 day to 30 years of age, containing 234 subjects (median 9 years, IQR 0.45–14 years). Whole-brain averaged EEG spectra are shown in Supplementary Fig. 1a, b. The canonical emergence and increase in frequency of the posterior dominant rhythm (PDR) with advancing age is apparent in subjects older than one year, as are changes in spectral slope.

### Tuning SGM parameters generates spectral shifts and the appearance of PDR

Next, we evaluated whether the SGM recapitulates spectral changes in periodic and aperiodic components of EEG spectra by systematically varying SGM parameter values to generate different spectral output realizations. Specifically, we evaluate the PDR and spectral slope, which reflects the $1/f$ distribution of aperiodic power across all frequencies[18]. Figure 2a demonstrates the emergence of the PDR with increasing values of $\alpha$, mirroring the monotonic increase of PDR peak frequency that is seen with typical neurodevelopment. There is also an increase in the spectral slope with increasing values of $\alpha$. Figure 2b demonstrates an increasing spectral slope in the alpha to beta frequency range with increasing values of $G_{EI}$. Figure 2c demonstrates a slight increase in the PDR with increasing conduction speed while the slope of the power spectra remains relatively unchanged. Figure 2d demonstrates the emergence of PDR with increasing values of $G_{II}$. There is also an increasing spectral slope of the alpha to beta frequency range with increasing values of $G_{II}$. These changes in spectral slope generated by changes to SGM input parameters suggest that SGM is suitable for capturing changes in aperiodic $1/f$ activity seen with EEG maturation[19-21].

### UMAP analysis demonstrates SGM spectral similarity to empirical EEG data and parameter space indeterminacy

Expanding on the finding that SGM parameter variation replicates key EEG developmental spectral trajectories, we next examined the fidelity of SGM simulations to actual, empirically observed EEG spectra. Utilizing Uniform Manifold Approximation and Projection (UMAP) for dimensionality reduction of high dimensional spectra, we juxtaposed the low-dimensional latent representations of SGM simulated spectra and observed spectra. The resulting UMAP embeddings are shown in Fig. 2e, f and Supplementary Fig. 2, with a total of 196,000 SGM realizations displayed. Supplementary Fig. 2 demonstrates an overlap between the simulated SGM and observed EEG spectral embeddings, indicating that SGM parameter variation replicates real-world EEG spectral features observed across development. Figure 2f, g demonstrates regions with homogeneous PDR frequency or aperiodic exponent values within the UMAP EEG spectral embedding space. This indicates model indeterminacy or regimes in SGM parameter space that yield similar spectral features.

### Effects of structural connectome on SGM power spectral density

We utilized the template Human Connectome Project (HCP) structural connectome[40] in preceding parameter variation analysis and subsequent analyses due to the unavailability of age-specific structural connectomes across the broad pediatric age ranges and fine temporal resolutions (weeks) required for our study. To assess the effect of excluding age-dependent connectomes, we contrasted SGM realizations derived from structural connectivity at the neonatal and adult developmental extremes. As macroscopic spectral activity in the SGM is determined by the long-range coupling ($\alpha$) of localized neural dynamics propagated across the structural connectome, we evaluated the effect of utilizing neonatal versus adult group-averaged structural connectomes on SGM spectral realizations at both strong and weak $\alpha$ regimes. The selection of structural connectome did not significantly alter spectral power distribution (Supplementary Fig. 3). In contrast, using strong $\alpha$, relative to weak $\alpha$, in the SGM increased spectral power and markedly altered the spectral power distribution, leading to emergence of PDR (Fig. 3). There was a significant difference in the spectral distribution shift induced by $\alpha$ selection compared to the shift induced by structural connectome selection, with a whole-brain Jensen–Shannon divergence difference of 0.0899, $p < 0.001$. These findings suggest that the contribution of structural connectomes in shaping the generation of $1/f$ activity and PDR is limited, corroborating recent evidence from the developing Human Connectome Project (dHCP) indicating that core structural connectome components are established in utero and stable postnatally[41]. Therefore, although the utilization of a static structural connectome does not encompass all structural network changes affecting spectral maturation, it nonetheless offers a foundation to understand how spectral maturation may emerge from age-dependent tuning of localized neuronal dynamics and their long-range coupling within a conserved structural network.

### SBI sensitivity analyses, posterior diagnostics, and simulation-based calibration

To elucidate key parameters driving EEG spectral development, we utilized a Bayesian approach employing SBI to identify approximate posterior distributions of SGM parameters that best align with synthetic or empirical EEG spectra. We evaluated three recent SBI methods: Neural Ratio Estimation (NRE)[42], Automatic Posterior Transformation, also known as Neural Posterior Estimation (NPE)[43], and the recently introduced truncated sequential NPE (TSNPE), which more robustly handles posterior estimation at the boundaries of the specified prior range[44]. In order to assess the accuracy and robustness of the SBI-SGM inference framework, we conducted parameter recovery analyses and simulation-based calibration (SBC) with synthetic data as outlined in Supplementary Note 1.

NRE, NPE, and TSNPE demonstrated differential performance across parameter recovery and SBC, with NPE and TSNPE generally outperforming NRE (Supplementary Figs. 4–11). While NPE demonstrated robust parameter recovery, it tends to produce poorly calibrated posterior

## Table 1 | Spectral graph model parameters and bounds

| Parameter | Symbol | Bounds |
|---|---|---|
| Excitatory time constant | $\tau_e$ | [0.001, 0.03] |
| Inhibitory time constant | $\tau_i$ | [0.001, 0.03] |
| Long-range coupling constant | $\alpha$ | [0.01, 1.0] |
| Conduction speed | $S$ | [0.5 m/s, 15 m/s] |
| Excitatory gain | $G_{EI}$ | [1, 20] |
| Inhibitory gain | $G_{II}$ | [1, 20] |
| Graph time constant | $\tau_G$ | [0.01, 0.3] |

**a**

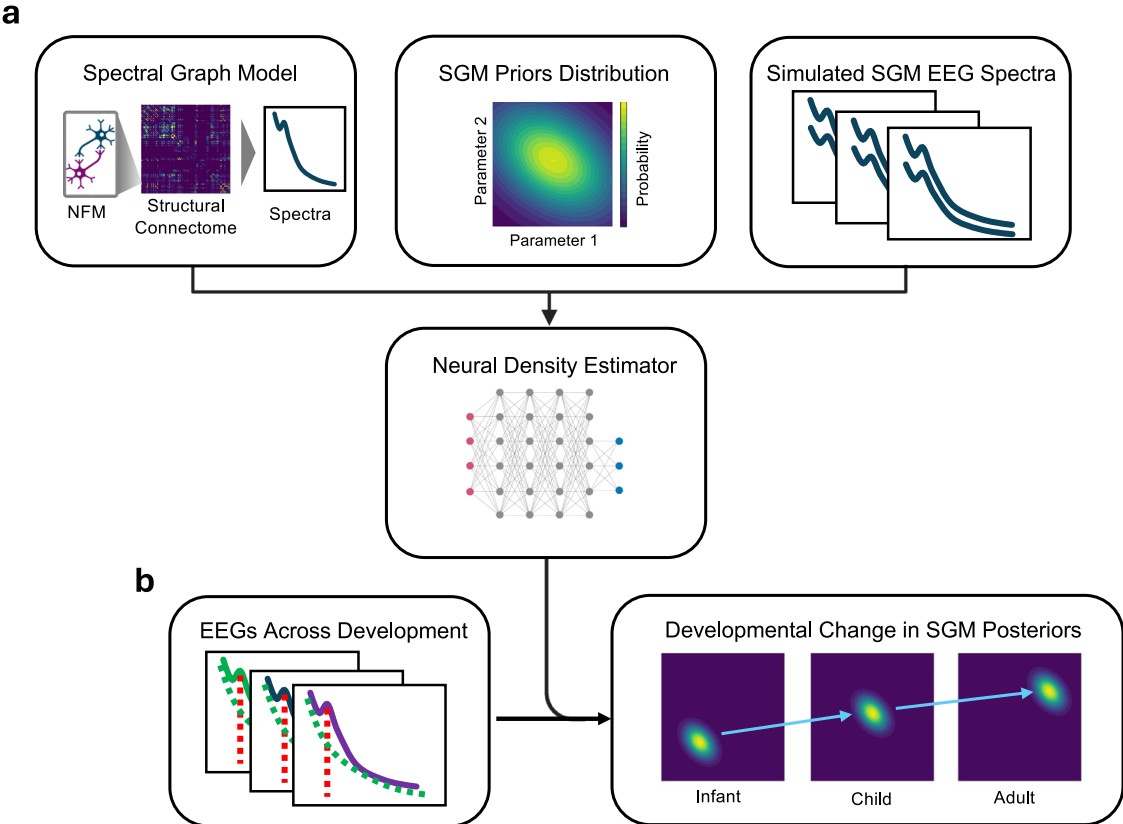

**Fig. 1 | Study overview.** We utilize simulation-based inference (SBI) with neural density estimation to approximate age-varying Spectral Graph Model (SGM) parameter posterior distributions from EEG spectra spanning the developmental period. **a** During Neural Density Estimator (NDE) training, the NDE learns the mapping between simulated SGM EEG spectra and their summary statistics realized from SGM parameterizations sampled from the SGM prior distribution. **b** SBI with the trained NDE is then used to infer approximate SGM parameter posterior distributions from subjects at different ages across development. Representations of hypothetical infant, child, and adult EEG spectra examples representing developmental EEG changes are shown. Example summary statistics that are provided as input to the NDE include posterior dominant rhythm (PDR) center frequency (red-dashed line), aperiodic exponent (green-dashed line), and the spectral distribution (green, blue, and purple lines for infant, child, and adult examples). Subsequent inferred approximate posterior distributions for respective hypothetical infant, child, and adult examples demonstrate developmental shifts (blue arrows) in SGM parameters. NFM Neural field model.

distributions (Supplementary Fig. 12) and is prone to posterior leakage at the parameter bounds (Supplementary Fig. 13). In contrast, TSNPE has well-calibrated posteriors relative to NPE and is robust against posterior leakage (Supplementary Fig. 13); however, its scalability is limited under conditions of extensive simulation requirements and large datasets. NPE and TSNPE captured similar correlation structures in the estimated posterior joint marginal distributions (Supplementary Fig. 14). We discuss potential sources of the differential performance across SBI methods further in the Supplementary Note 2. Given the general preference for conservative over overconfident posterior estimates—the latter potentially leading to erroneous scientific conclusions[45]—we utilize TSNPE in the subsequent application of SBI-SGM to empirical data given its improved calibration results compared to NPE.

### SBI of SGM parameters recapitulates observed EEG spectral features

Having assessed the robustness of the inference procedure, we next performed a posterior predictive check on empirical data, specifically simulating spectra under the fitted SGM and then comparing these to the observed data. Figure 4 (red traces) demonstrates examples of output SGM spectra realized from mean values of the SGM parameter posterior distribution inferred with TSNPE that resemble the input empirical EEG spectra (black traces). These examples recapitulate periodic and aperiodic components of the empiric EEG spectra, confirming that SBI with SGM captures relevant spectral features of observed EEG data, including periodic

and aperiodic features such as PDR and spectral slope. The corresponding posterior distributions for each respective EEG spectrum SGM model fit with TSNPE are shown in Fig. 4 (right panels), with higher probabilities assigned to SGM parameter sets that generate realizations consistent with the observed data and lower probabilities to inconsistent parameter sets. NPE and TSNPE had similar Posterior Dispersion Index (PDI) profiles for empirical data, with both demonstrating excitatory and inhibitory time constants as having the least dispersion and conduction speed as having highest the dispersion (Supplementary Fig. 11).

### SBI of SGM parameters shows age-related progression in long-range coupling and axonal conduction speed, and excitatory: inhibitory balance

Next, we aimed to ascertain if age-dependent trajectories in the SGM parameter posterior distributions, inferred from real EEG data, would align with established age-dependent changes in their biological counterparts. These developmental trends include well-known increases in functional long-range connectivity and coupling strength[46,47], acceleration of axonal conduction speed[48], and shift towards reduced excitatory:inhibitory balance with aging[49–51]. To achieve this, we allowed the SBI-SGM inference to operate within physiologically plausible prior bounds, with no other enforced constraints, thereby allowing model flexibility within a broad parameter space. This approach allowed us to evaluate the natural evolution of SGM parameter posterior distributions across the developmental time frame.

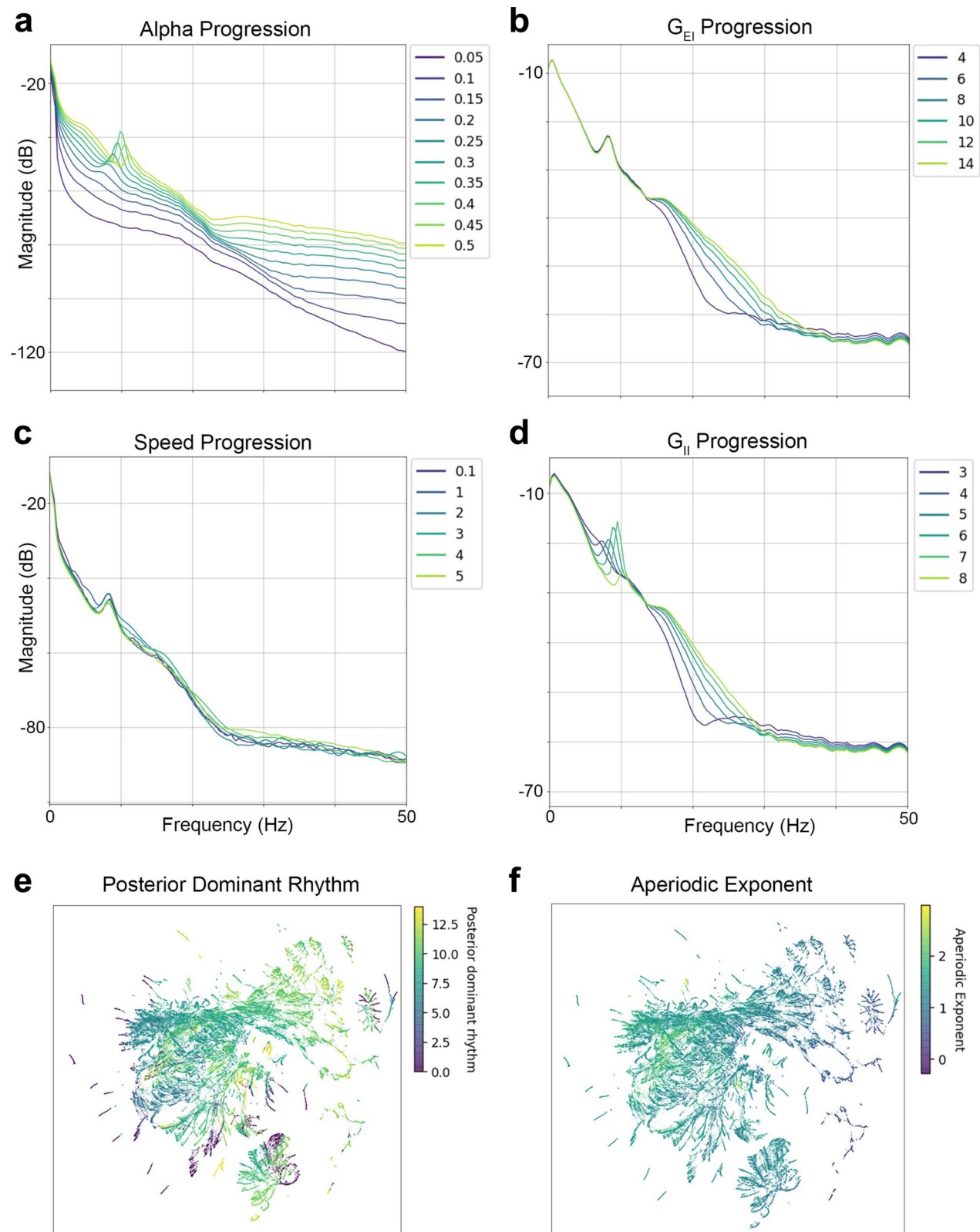

**Fig. 2 | Spectral realizations with varying SGM parameterization. a–d** demonstrates the effect of varying Spectral Graph Model (SGM) parameters. **a** $\alpha$ parameter progression shows emergence and increase in the frequency of posterior dominant rhythm as $\alpha$ is increased, **b** Excitatory gain progression shows a change in aperiodic content of spectral frequencies greater than 10 Hz. **c** Axonal conduction speed parameter progression shows an increase in PDR frequency as speed (m/s) is increased. **d** Inhibitory gain progression shows the emergence of posterior dominant rhythm (PDR) and increase in PDR frequency as well as changes in the aperiodic content of spectral frequencies greater than 10 Hz. **e** Uniform Manifold Approximation and Projection (UMAP) embedding showing gradients in PDR frequency (Hz) evident in different clusters. Spectra without PDR are shown in purple. There are regions with homogenous PDR frequency suggesting SGM parameter regimes that yield similar PDR. **f** UMAP embedding showing gradients in aperiodic exponent evident in different clusters. There are regions with homogenous aperiodic exponents suggesting regions of SGM parameter space that yield similar aperiodic exponents.

**Fig. 3 | Effects of neonatal versus adult connectome and coupling strength on SGM spectral realizations.** We compared the differential effects of utilizing a neonatal versus adult connectome and strong versus weak long-range coupling ($\alpha$) on Spectral Graph Model (SGM) spectra realizations across different brain regions (frontal, temporal, parietal, occipital, and whole-brain). The left and right columns demonstrate SGM power spectral density (PSD) realizations instantiated with group-averaged neonatal and adult connectomes, respectively ($N = 1000$ per connectome). Each subplot shows mean SGM realizations instantiated with weak (red-dotted line) and strong (blue line) $\alpha$, with 95% confidence intervals (CI) indicated by the corresponding shaded regions. $\alpha$ was sampled uniformly at random between 0.1 to 0.3 for weak and between 0.7 to 0.9 for strong $\alpha$ regimes, respectively. Remaining SGM parameters were uniformly randomly sampled from physiologically-informed prior ranges (Table 1). Qualitatively, there is a broadband increase in spectral power with relatively stronger augmentation in alpha power seen with stronger $\alpha$. The alpha peak is not as well defined as seen in Fig. 2 due to the effects of PSD averaging. There were subtle differences in PSD distribution resulting from selection of neonate versus adult structural connectome, demonstrated in Supplementary Fig. 3. Distances between normalized mean spectra per connectome were assessed with Jensen–Shannon divergence (JSH). Across all cortical regions, there was a significant difference in the spectral distribution shift induced by $\alpha$ compared to the shift induced by structural connectome, with whole-brain JSH difference of 0.0899 ($p < 1.0\mathrm{e}{-7}$).

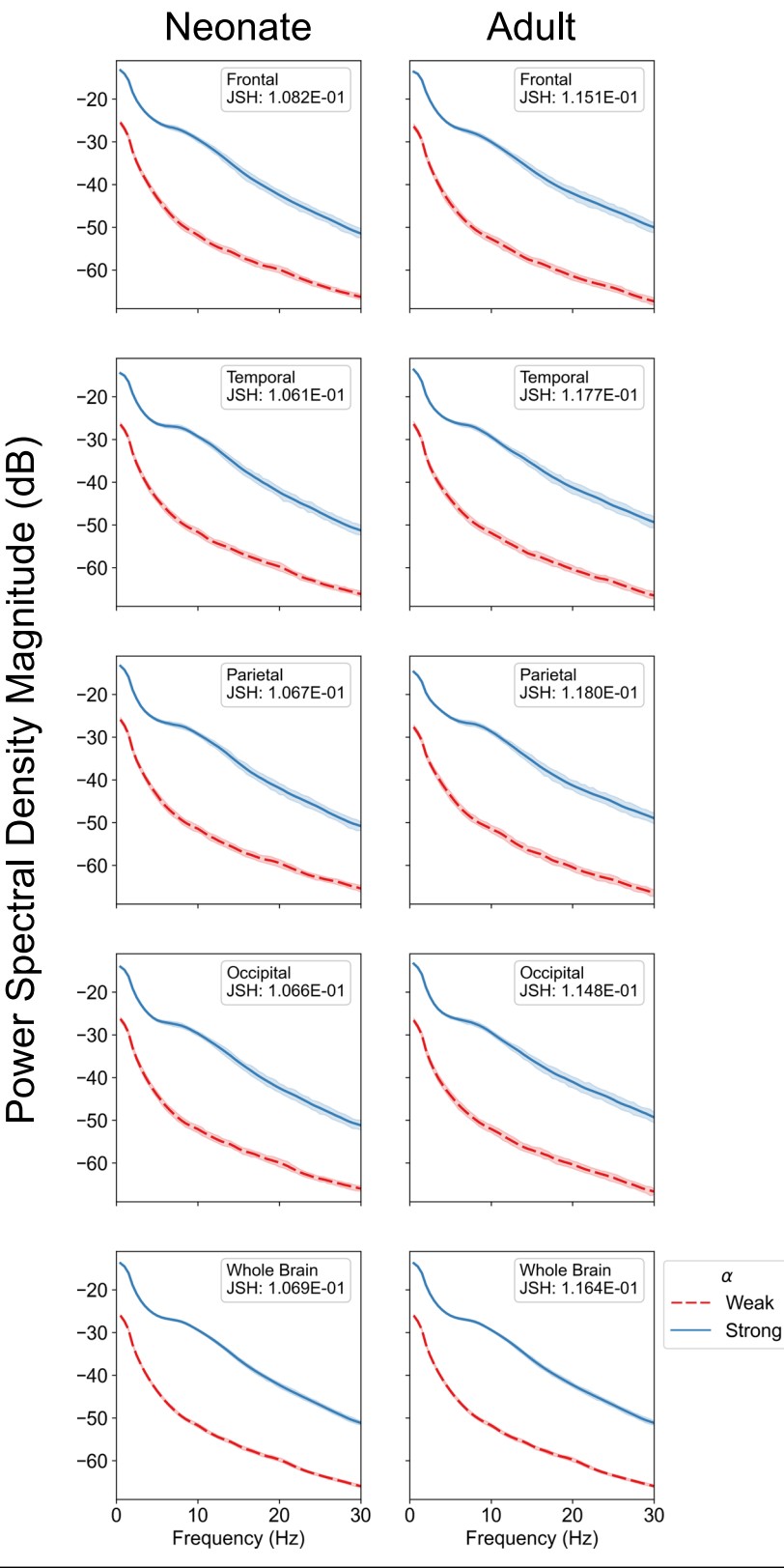

We inferred multivariate SGM posteriors for each subject with TSNPE (Supplementary Figs. 15 and 16), then retrieved the posterior means from the probability density function for each respective SGM parameter posterior distribution. Figure 5 demonstrates plots of the subsequent mean SGM parameter values versus the respective age for each subject. There was a significant positive association between age and long-range coupling $\alpha$ (Pearson correlation coefficient $r = 0.615$,

$p < 0.001$. Considering that $\alpha$ reflects functional long-range coupling of neural dynamics unfolding over a seconds-scale timespan[52], the protracted increase in PDR occurring over years during neurodevelopment suggests a gradual enhancement in the baseline tone of long-range functional coupling within the connectome. Biologically, this may be mediated by the strengthening of long-range excitatory and GABAergic coupling that occurs during neurodevelopment[46,47].

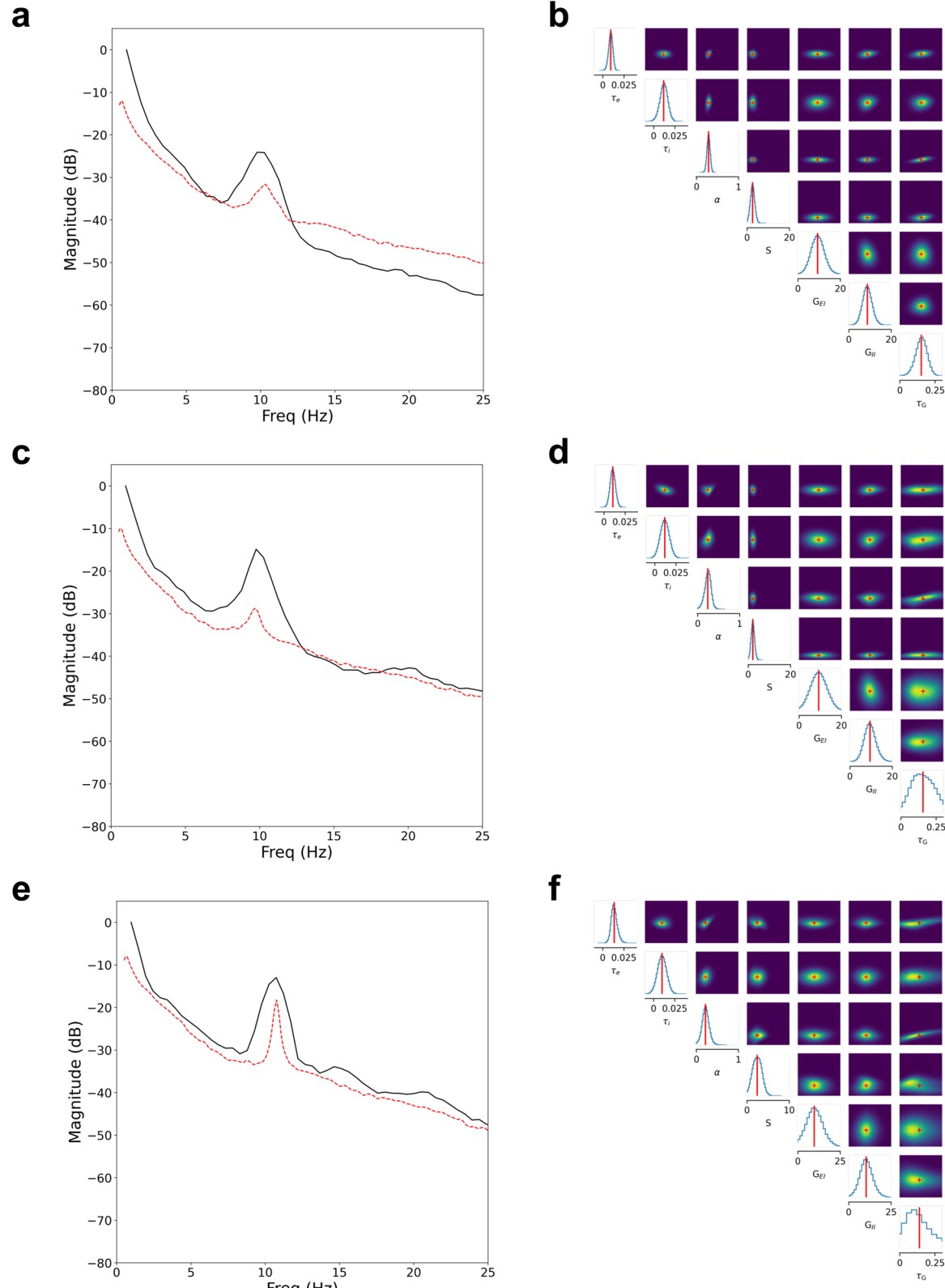

**Fig. 4 | Observed subject EEG spectra, inferred SGM parameter posterior distributions, and corresponding simulated spectra.** Posterior predictive checks, comparing simulation-based inference (SBI) of the spectral graph model (SGM) derived spectral realizations to observed data, are shown for three representative subject EEGs **a**, **c**, **e**. Observed EEG spectra are demonstrated in **a**, **c**, **e** (black traces), and their respective approximated SGM parameter posterior distributions obtained with SBI are shown in **b**, **d**, **f**. The mean values (red lines in **b**, **d**, **f**) of respective SGM parameter posterior distributions were used to simulate spectra with SGM and yielded simulated spectra (red-dashed traces) shown in **a**, **c**, **e**. The simulated spectra have characteristics similar to those of the observed data, including posterior dominant rhythm (PDR) center frequency and spectral slope.

**Fig. 5 | SBI of SGM parameters over the developmental period.** The evolution of inferred spectral graph model (SGM) parameters is visualized over time. Each colored circle represents the peak value of the respective SGM parameter distribution (*y*-axis) for a respective subject, as inferred by simulation-based inference (SBI), plotted over respective age of the subject (*x*-axis). The linear regression model fit is represented by the solid line and the shaded area represents 95% confidence interval. α demonstrated Pearson $r = 0.284$ ($p = 3.88e-5$). Axonal conduction speed demonstrated $r = 0.350$ ($p = 2.97e-7$). Excitatory Gain demonstrated $r = -0.430$ ($p = 1.40e-10$). Inhibitory Gain demonstrated $r = 0.388$ ($p = 1.01e-5$). Graph time constant did not demonstrate a linear correlation with age (Supplementary Fig. 17). The *x*-axis tick labels are log-scaled for visualization purposes. Symbols and abbreviations: Excitatory time constant, $\tau_e$ (blue); Inhibitory time constant, $\tau_i$ (orange); Long-range coupling constant, α (green); Axonal conduction speed, *S* (red); Excitatory gain, $G_{EI}$ (purple); Inhibitory gain, $G_{II}$ (brown).

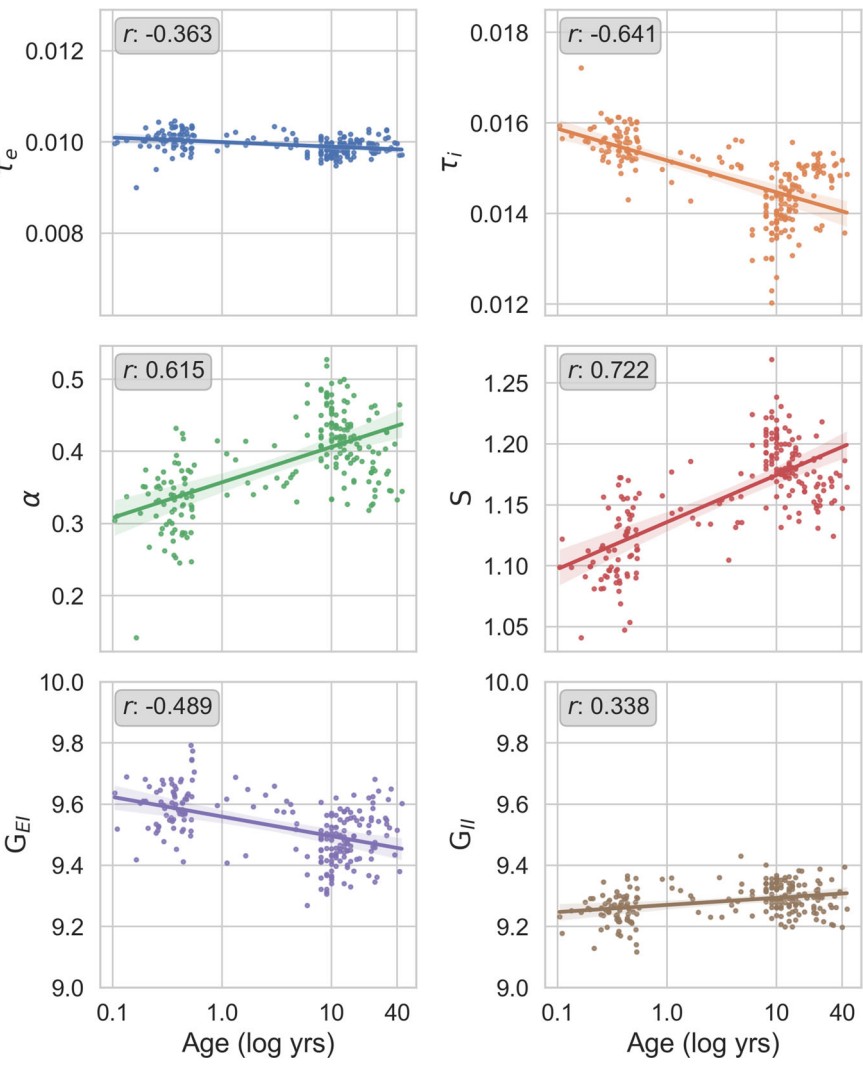

In addition, our results indicate an age-dependent increase in axonal conduction speed *S* ($r = 0.722$, $p < 0.001$), consistent with the known maturation of white matter pathways, which also importantly impacts long-range coupling[48]. This finding is congruent with the established maturation of white matter myelination and axon diameter, which are the primary determinants of conduction speed[53,54].

Furthermore, we demonstrate an age-dependent reduction in excitatory:inhibitory (E:I) balance ($G_{EI}$: $r = -0.489$, $p < 0.001$; $G_{II}$: $r = 0.388$, $p < 0.001$), mirroring animal model studies that have reported a developmental decrease in the E:I balance, putatively driven by an increase in GABAergic tone[55,56]. Also, recent human studies have investigated TMS to measure and affect the E:I balance[49,50], and the finding of reduced E:I during development aligns with TMS study findings that older individuals tended towards greater intrinsic inhibitor tone[51].

Lastly, $\tau_e$ and $\tau_i$ demonstrated negative associations with age with $r = -0.363$ ($p < 0.001$) and $r = -0.641$ ($p < 0.001$), while there were no associations between age and $\tau_G$ (Supplementary Fig. 17). These reductions in $\tau_e$ and $\tau_i$ are potentially congruent with rodent and primate studies, which have reported a general quickening of GABAergic and glutamatergic synaptic time constants over the developmental period[57–61]. However, regression diagnostics with scale-location plots (Supplementary Fig. 18) for $\tau_e$ and $\tau_i$ indicated heteroskedasticity confirmed by Breusch-Pagan tests with $\chi^2$ of 5.560 and 14.13 with corresponding *p*-values of 0.0179 and 1.71e−3, respectively. The presence of significant heteroskedasticity limits the reliability and validity of the $\tau_e$

and $\tau_i$ findings. Regression diagnostics on the other SGM parameters did not exhibit significant heteroskedasticity.

## Prediction of age and PDR with data-driven inference of SGM parameters

For further validation of the age-dependent trajectories of the inferred SGM parameter posterior distributions, we next asked whether the inferred SGM parameters could predict subject ages from the input EEG spectra. Employing polynomial regression, we modeled the relationship between SGM parameters that demonstrated robust linear associations with age (α, *S*, $G_{EI}$, $G_{II}$) with log age. We evaluated the predictive capacity of the regression model on observed versus predicted age using a cross-validation approach and compared performance to a polynomial regression model fit on periodic and aperiodic parameters obtained with the *fitting oscillations & one over f* (FOOOF) methodology[18]. We evaluated for heteroskedasticity with the Breusch-Pagan test and the SBI-SGM regression model did not exhibit heteroskedasticity (Supplementary Fig. 19). The adjusted coefficient of determination ($R^2$) for observed versus predicted age was 0.534 (Fig. 6a), consistent with good agreement between SBI-SGM regression model predicted age and observed age. This demonstrates improved performance over the FOOOF regression model, which had ($R^2$) of 0.534 (Fig. 6b). We then evaluated whether inferred SGM parameters would correspond with center frequency changes observed in the PDR over time. We modeled this relationship using polynomial regression and found an adjusted $R^2$ of 0.217 for observed versus predicted PDR (Fig. 6c), suggesting that the evolution of SGM features during development only partially accounts for the observed

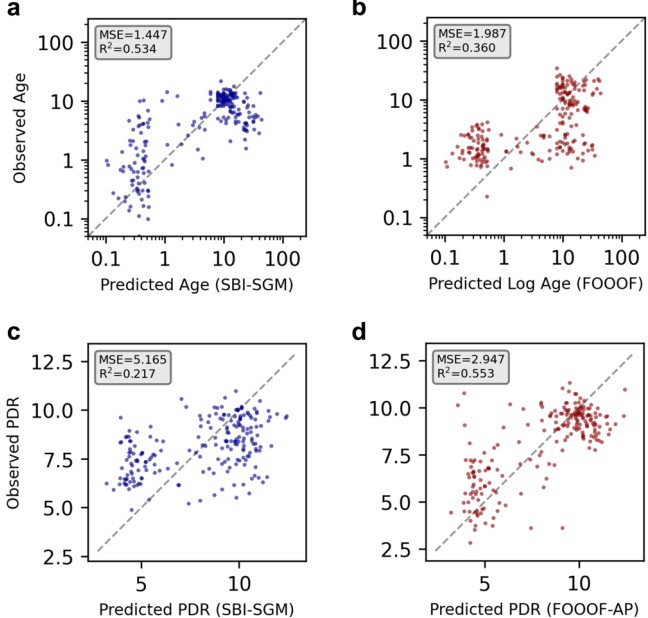

**Fig. 6 | Prediction of age and PDR from inferred SGM parameters. a** Observed versus predicted age plot for predicted ages derived from linear regression model utilizing simulation-based inference of Spectral Graph Model parameter posterior distribution means (SBI-SGM). **b** Observed versus predicted age plot for predicted ages derived from linear regression model utilizing posterior dominant rhythm (PDR) bandwidth, PDR center frequency, PDR power, aperiodic exponent, and aperiodic offset obtained with the Fitting Oscillations & One Over F (FOOOF) method. Respective Adjusted $R^2$ and mean square error (MSE) values are shown in the gray subset boxes. SBI-SGM demonstrated improved prediction of age compared to FOOOF, with relatively higher $R^2$ and MSE compared to FOOOF. **c** Observed versus predicted PDR plot for PDR derived from linear regression model utilizing SBI of SGM parameter posterior distribution means (SBI-SGM). **d** Observed versus predicted PDR plot for predicted PDR derived from linear regression model utilizing aperiodic exponent and offset obtained with FOOOF (FOOOF-AP). FOOOF-AP demonstrated improved prediction of the PDR compared to SBI-SGM.

variance in periodic EEG spectral features. In comparison, the FOOOF model fit on aperiodic $1/f$ exponent and intercept achieved $R^2$ of 0.553, suggesting that aperiodic $1/f$ features evolution during aging are predictive of PDR frequency. Prior literature in predicting age from EEG has reported $R^2$ values ranging up to 0.61 for datasets including pediatric subjects, with the best results achieved utilizing machine learning approaches[62]. Engemann et al. reported deep learning (DL) and feature-engineering models had R2 values of 0.61 and 0.33, respectively, on the normal subgroup of the Temple University Hospital Abnormal (TUAB) EEG dataset ($N = 1385$, mean age: 48.6 years with standard deviation 17.9 years)[62]. Given that TUAB predominantly consists of adult subjects with 43 subjects between 10–20 years and only three subjects in the 0–10 years, direct comparison with our findings, which focus on the developmental age demographic, is limited. In contrast to DL or feature engineering approach utilized by Engemann et al., the latter of which utilized at least 30 features, the SGM approach demonstrates comparable performance while utilizing a biologically interpretable feature set with 7 parameters, thus providing mechanistic insight into the underlying brain dynamics correlated with aging and development.

## Discussion

In this study, we utilized SBI with the spectral graph model (SBI-SGM) to model EEG spectral maturation and infer population-based trajectories of SGM parameters relevant to brain development, such as long-range coupling and excitatory-to-inhibitory (E:I) balance. We demonstrate that the temporal progression of SGM parameters coheres with their expected developmental evolution. To the best of our knowledge, this is the first

demonstration that the maturation of brain spectra across developmental stages can be accurately modeled within an analytical framework, guided by the neurobiologically consistent evolution of model parameters.

The maturation of long-range projection fibers, marked by myelination and axonal diameter growth, are considered crucial in promoting structure:function coupling thereby mediating the evolution of diverse functional network configurations and their underlying brain rhythms[13,63–67]. However, recent evidence indicates that the connectome's structural core is already present and remarkably stable in infancy and early childhood[41]. This suggests the presence of additional mechanisms contributing to the postnatal refinement of functional networks.

In exploring these mechanisms with SGM-SBI, we identified that an increase in long-range coupling ($\alpha$) during development captured the canonical age-dependent increase in the PDR. While PDR and other oscillations can be generated via neural mass models modeling cortical columns or the thalamocortical system[68–72], to our knowledge how certain brain oscillations such as PDR may emerge and evolve at expected time points in the lifespan has not been previously specified with a mechanistic whole-brain model. Our findings suggest that PDR evolution is partially driven by age-dependent alterations in network dynamics underpinned by the gradual strengthening of long-range functional coupling over the structural connectome[47]. Also, our results recapitulated the known increase in axonal conduction speed with age, which facilitates transmission efficiency and thus functional organization of long-range neural networks[48,73]. Finally, we demonstrated that E:I balance demonstrated a reduction with age, congruent with the established reduction of E:I balance that occurs with development, thought to be primarily mediated by the maturation of GABAergic circuitry[74–76]. Fine-tuning of E:I balance has been demonstrated across different brain regions and disruptions to E:I balance have been implicated in animal models of autism and other disorders of disrupted neurodevelopment[6,7,55,77]. We assumed uniform levels of excitation and inhibition, respectively, occurring at a mesoscopic level throughout the whole brain network model of the SGM. While this is a simplified treatment of E:I balance, we recognize that excitation and inhibition are, in fact, multidimensional entities, occurring across multiple network scales, varying time periods, and mediated by diverse excitatory and inhibitory cell types and synaptic mechanisms[7]. Nonetheless, SGM provides an analytical toehold in the challenge of understanding E:I balance by relating changes to configurations of excitatory or inhibitory gains to predicted brain spectral activity[38].

Spectral graph theory, and in particular one of its central tenets the graph Laplacian operator, has broad applicability and effectiveness across multiple domains in network science[78], thus has emerged as a powerful tool for understanding structure-function relationships in neuroscience. Indeed, eigenmodes of the brain network Laplacian, as introduced by Abdelnour et al. and others, have been used to predict canonical functional networks from underlying structural networks[79–81]. These models are intimately related to walks on graphs, and higher-order walks on graphs have also been quite successful; typically, these methods involve a series expansion of the graph adjacency or Laplacian matrices[82,83]. It is now known that the series expansion and eigenmode approaches yield highly similar mappings between functional and structural networks[84–86]. In the latter, eigenmodes of the adjacency or Laplacian matrix are typically employed, and only a few eigenmodes are usually sufficient to reproduce empirical functional connectivity[84,87–92]. While the above eigenmode-based models have demonstrated the ability to capture steady-state, stationary characteristics of real brain activity, they are limited to modeling passive spread without oscillatory behavior. Capturing the rich repertoire displayed by EEG recordings would require a full accounting of axonal propagation delays as well as local neural population dynamics within graph models. Band-specific MEG resting-state networks were successfully modeled with a combination of delayed neural mass models and eigenmodes of the structural network[93], suggesting that delayed interactions in a brain's network give rise to functional patterns constrained by structural eigenmodes. The SGM further develops this concept by demonstrating the prediction of

spatial and spectral features of neural oscillatory activity[38], providing a parsimonious and principled framework for modeling electrophysiologic maturation during development.

Next, we contrast our study to prior approaches to modeling whole brain activity. While there have been numerous large-scale neural network or macroscopic brain models of alpha frequency rhythms (PDR) proposed, these have been generally limited to modeling alpha activity in relatively more temporally constrained brain states, such as during anesthesia[68,70,71,94]. While early approaches successfully captured oscillatory brain activity[68,70,71], they did not aim to simultaneously model aperiodic brain activity. Conversely, Bedard and Destexhe developed a framework for the genesis of aperiodic brain activity; however, this framework did not capture periodic oscillations[95]. More recently, Hashemi et al. utilized Bayesian inference techniques, including Monte Carlo Markov Chain methods, to effectively fit a thalamocortical neural mass model to EEG data in humans undergoing anesthesia, demonstrating effective capture of key spectral peaks in delta and alpha frequencies, as well as aperiodic components, in observed data[96]. Few studies have appeared to utilize a regional or whole-brain network approach to seek to explain spectral periodic and aperiodic activity. An approach utilizing the Kuramoto model with a whole brain connectome suggested that mechanistic underpinnings of alpha and beta oscillatory activities arise from cross-regional synchronization; however, de-emphasized the accuracy of their simulation at capturing realistic brain activity[72]. Our finding of the importance of long-range coupling ($\alpha$) in the emergence of alpha-range oscillatory activity (PDR) aligns with this perspective, and moreover, offers a neurobiological mechanistic underpinning for how increased network synchronization or long-range coupling gives rise to oscillatory activity.

Subsequently, we address that the stability or robustness of our findings hinge on several aspects: data characteristics and degree of model misspecification; SBI components such as prior specification and posterior approximator hyperparameterization[97]; and the stability of the SGM itself. Regarding dataset characteristics, the experimental data distribution, noise levels, and alignment (or misalignment) of the model with the data-generating process may affect SBI robustness[98]. EEG data demonstrates substantial variability across the developmental timeline. Thus, depending on the age group of interest, a more targeted analysis within narrower developmental windows may yield more robust inferences due to decreased EEG variability and more informative, age-specific priors. Concerning stability considerations related to the SBI, our evaluations utilizing synthetic reference data demonstrated that stability and accuracy of posterior distributions, given a priori knowledge of the true solution, depended on NPE hyperparameterization, such as the simulation budget utilized. In addition, one may consider approaches to improve SBI stability and reliability, including ensembling and post-hoc calibration methods[99]. Further work involving generalization studies on larger datasets is required to effectively evaluate the stability of the SBI-SGM framework. Regarding numerical stability of the SGM model itself, Verma et al. applied root locus analysis to delineate the bounds of SGM parameters that give rise to dynamical behaviors, including damped oscillations, limit cycles, or unstable oscillations[52]. Our SBI findings for excitatory and inhibitory time constants and alpha were within ranges that promote stable oscillatory activity. In contrast, excitatory:inhibitory gains were consequently inferred to be above boundaries that ensure stable oscillations, suggesting a potential for instability. These parameters exhibited increased posterior dispersion and variance in identifiability, which could reflect the unstable regime in SGM parameter space leading to greater unpredictability in SGM output. However, our empirical EEG data encompassed subjects who exhibited fluctuating and unstable oscillatory activity that was interspersed with prominent $1/f$ activity, or lacked oscillatory activity altogether as in the case of younger infants. The ability of the SGM to account for such diverse neural dynamics promotes robustness in the SBI-SGM framework, particularly in accommodating the unpredictability and instabilities of whole-brain network activity.

We acknowledge several limitations in our study. A primary limitation arises from the non-utilization of age-specific structural connectomes from birth to adulthood; rather, analysis was performed using a standardized adult connectome. This constraint is primarily due to scarcity of available age-specific structural connectome datasets for the 0 to 24 month old age range with fine-grained age resolution. Consequently, our analysis does not account for potential changes in the structural connectome that may contribute to spectral maturation. Recent evidence suggests that the structural connectome forms in utero[100,101]. However, our finding that the utilization of neonatal versus adult structural connectome had relatively subtle effects on SGM spectral realizations aligns with recent findings from Ciarrusta et al., who demonstrated that the core components of the structural connectome develop in utero and are relatively stable postnatally[41]. In addition, the functional connectome fingerprint is not only established, but also demonstrates stability during early brain development in infants[102]. Despite the limitations arising from the use of a static structural connectome in our analysis, maintaining a stable structural network over the developmental age range in our analysis provides insight into how functional changes arise upon a consistent structural framework.

Furthermore, we acknowledge a potential limitation in our analysis arising from relatively unrestricted parameter exploration, constrained only by bounds on SGM time constants and propagation parameters. During SBI and within the SGM, we did not enforce neurobiological constraints, thus the inferred SGM parameter realizations may not necessarily align with naturally observed relationships in the parameters. Hashemi et al. utilized neurobiological constraints enforced during spectral fitting, such as differential response function characteristics for excitatory and inhibitory synapses with excitatory response function having longer rise and decay times than inhibitory synapses[96]. Within the proposed SBI-SGM framework, we purposefully did not enforce neurobiological constraints with the aim of capturing a parsimonious description of brain spectral evolution over development. Nevertheless, introducing such constraints would enable further alignment of the SBI of brain and neural models with naturally occurring observed constraints.

Also, we note that the SGM is inherently a linear model, and thus may be limited in its ability to fully capture all the complexities of nonlinear neural dynamics[103]. However, despite the intrinsic nonlinearities present in the brain, particularly at microscopic and mesoscopic scales, a recent study comparing linear and nonlinear approaches of modeling macroscopic intracranial EEG and fMRI neurophysiologic activity demonstrated that linear models unexpectedly performed more accurately than nonlinear models, thus suggesting potential advantages to the linear modeling approach beyond their relative interpretability compared to nonlinear models[104]. However, it is crucial to recognize that the intrinsic nonlinearities and complexities of brain network dynamics may be more faithfully represented through nonlinear modeling[103]. Thus, the advantageous interpretability and computational tractability of linear models such as the SGM should be seen as a complement to, rather than a replacement for, nonlinear methods.

Additionally, we recognize the argument that fitting our empirical data, constituting an average EEG spectrum across channels, may not necessitate a spatially resolved model like the SGM. In fact, one could fit the empirical spectrum to a single lumped neural mass model[69]. More detailed models are also available[105]. Such models would incorporate local dynamic properties like E:I gain and local circuit time constants. However, they would be unable to probe the specific maturation behavior of interest seen developmentally - i.e., of axonal conduction speed and inter-regional coupling. Another benefit of a spatially resolved model such as the SGM is that in future studies with denser EEG configurations or scalp MEG, it would be possible to perform reliable source reconstruction, which will in turn enable the interrogation of the spatial gradients of empirical electrophysiology, in addition to their spectral content. This would, for instance, allow us to explore the ability of fitted model to reproduce the spatial dominance of alpha rhythm, analogous to that recently demonstrated in adults[38,92].

Lastly, we observe that the ability of SBI capture age-dependent changes in SGM parameters from empirical EEG data suggests broader applicability. For instance, in epileptogenesis, elucidating the time-dependent slow dynamics of long-range coupling and excitatory:inhibitory balance may provide insights into pathologic neurophysiological shifts that engender the development of epilepsy, whereas identifying rapid changes in these parameters could elucidate dynamics of pre-ictal and ictal states. Similarly, deviation of these parameters from typical trajectories could be used to investigate spectral differences that arise in autism and other neurodevelopmental disorders. Lastly, Lavanga et al. recent utilized SBI in conjunction with whole brain modeling to understand structure-function relationship underlying cognitive decline in aging[106]. Similarly, the SGM, which has been recently used to model abnormal neural oscillations and their cellular correlates in patients with Alzheimer's Disease[107], in conjunction with SBI offers an analytical framework to further investigate mechanisms underlying cognitive decline. The potential for SBI in conjunction with the SGM to provide novel mechanistic insights into both typical neurodevelopment and conditions with disrupted neurodevelopment, such as autism and epilepsy, will be evaluated in future work.

## Conclusions

While the critical trajectories of morphological and structural connectivity remodeling during development are well-established, the structure-function relationship remains to be clarified. Our findings suggest that the evolving structure-function interplay influences canonical features of brain spectral maturation, including the evolution of PDR and $1/f$ aperiodic activity. This interplay is inextricably linked to changes in brain morphology and structural networks, yet understanding the precise mechanisms underlying this connection warrants additional investigation.

## Methods

### Spectral graph model

The SGM is a linear model capable of simulating spatial and spectral patterns of macroscopic neural activity. In this hierarchical model of the brain's structure-to-function relationship, the macroscopic functional activity emerging from mesoscopic neural activity is summarized by a minimal set of global macroscopic parameters in a closed-form Fourier domain formulation. The simulated broadband spectrum and spatial patterns emerge from the information-rich contents of the brain's structural graph Laplacian. The key concepts of the macroscopic model will be highlighted here, while the detailed derivations are illustrated in the original publications[38,92].

**Complex graph Laplacian.** For a brain's white matter diffusion-derived structural network, its normalized graph Laplacian matrix characterizes the most probable paths a signal may take in a network. Here, we define the brain's structural connectivity matrix as $\mathbf{C} = c_{j,k}$, consisting of connection strengths between any brain regions $j$ and $k$. To incorporate the time delays caused by white matter streamline distances between brain regions, we utilize the properties of delay-induced phases in the Fourier domain and introduce a complex-valued connectivity matrix $\mathbf{C}^*(\omega) = c_{j,k} \exp(-j\omega\tau_{j,k}^v)$. Where delays $\tau_{j,k}^v$ is computed from the pairwise region distances divided by a constant speed $v$. This complex connectivity matrix not only incorporates distance-induced delays between nodes ($\tau_{j,k}^v$) into our network but also allows us to estimate network properties given a frequency of oscillation ($\omega$). Therefore, a degree normalized complex connectivity matrix at some frequency $\omega$ is defined as:

$$\mathbf{C}(\omega) = diag\left(\frac{1}{\mathbf{deg}}\right)\mathbf{C}^*(\omega) \qquad (1)$$

Where the degree vector $\mathbf{deg}$ is defined as $\mathbf{deg}_k = \sum_j \mathbf{c}_{j,k}$. The SGM propagates signals via the eigenmodes of the network's Laplacian matrix, and

a normalized Laplacian $\mathcal{L}$ of $\mathbf{C}(\omega)$ is defined as:

$$\mathcal{L}(\omega) = \mathbf{I} - \alpha\mathbf{C}(\omega) \qquad (2)$$

Where $I$ is the identity matrix and $\alpha$ is a global coupling constant parameter that weights the network connections.

**The macroscopic model.** The macroscopic model relies on the assumption that signal transmission between macroscopic brain regions is linear, and the changes in the signal's spectral contents can be summarized by linear filters. In the SGM, this macroscopic linear filter is a Gamma-shaped function $F(\omega) = \frac{\frac{1}{\tau^2}}{(j\omega + \frac{1}{\tau})^2}$. These linear filters in combination with the complex valued network Laplacian dictates the spectral and spatial spreading of signals from mesoscopic neuron assemblies:

$$\mathbf{X}(\omega) = \left(j\omega\mathbf{I} + \frac{1}{\tau_G}F(\omega)\mathcal{L}(\omega)\right)^{-1}\mathbf{H}_{local}(\omega)\mathbf{P}(\omega) \qquad (3)$$

In steady-state conditions, we assume the brain has uncorrelated Gaussian noise, therefore the driving function $\mathbf{P}(\omega) = \mathbf{I}$, where $\mathbf{I}$ is a vector of ones. The network level time constant $\tau_G$ parameterizes the macroscopic properties of the brain's structural network and region wise activity $\mathbf{H}_{local}(\omega)$ is transferred throughout the network via the characteristic paths set by the Laplacian $\mathcal{L}(\omega)$. The characteristic paths are computed by the eigendecomposition of $\mathcal{L}(\omega)$:

$$\mathcal{L}(\omega) = \mathbf{U}(\omega)\Lambda(\omega)\mathbf{U}^H(\omega) \qquad (4)$$

Where $\Lambda(\omega) = diag([\lambda_1(\omega), \ldots, \lambda_N(\omega)])$ is a diagonal matrix consisting of the eigen values of $\mathcal{L}(\omega)$. By incorporating the eigen modes $\mathbf{U}(\omega)$ into (3), we can show that the macroscopic frequency profile $\mathbf{X}(\omega)$ is:

$$\mathbf{X}(\omega) = \sum_i \frac{\mathbf{u}_i(\omega)\mathbf{u}_i^H(\omega)}{j\omega + \frac{1}{\tau_G}\lambda_i F(\omega)}\mathbf{H}_{local}(\omega)\mathbf{P}(\omega) \qquad (5)$$

An overview of the SGM is shown in Fig. 1a.

### UMAP of SGM simulations

To understand the SGM parameter space and evaluate the ability of the SGM to capture developmental EEG spectral shifts we utilized UMAP (Uniform Manifold Approximation and Projection) for dimension reduction of spectra and visualization of SGM simulated spectra and observed EEG spectra, using hyperparameters number of neighbors of 60 and minimum distance of 0.1[108]. This generates a low dimensional embedding of potential simulated SGM spectra and observed EEG spectra on which spectra that self-similar will cluster. Similarity between SGM spectra and observed spectra will manifest with overlapping clusters of these respective groups and conversely dissimilarity between the two groups will manifest as clusters with minimal or no overlap. To allow a balanced input to UMAP that is representative of the canonical EEG power bands, we provide as input to UMAP individual frequency binned power from 0.5 Hz to 12 Hz in 0.5 Hz bins, average power in the high alpha (12–20 Hz), beta (20–40 Hz), and gamma (40–55 Hz).

### Subject EEG and standardized structural connectome

We utilized publicly available neonatal and infant EEG datasets with age range 1 month to 1 year of age[109]. We also utilized a publicly available database of EEGs containing ages 5–18 years of age[110]. Additional EEG data were used from patients with normal EEG between the ages of 1 day to 5 years of age were retrospectively identified from the University of California San Francisco (UCSF) Epilepsy Monitoring Unit EEG Database or the UCSF Benioff Children's Hospital Neuro-Intensive Care Nursery database and selected for analyses with the following inclusion criteria: (1) Normal EEG at time of study and (2) No known history of seizures,

hypoxia-ischemic encephalopathy, stroke, or other known neurological condition at the time of study. All study procedures were approved by the institutional review board at UCSF. EEG for neonates utilized modified 10–20 system for neonatal subjects with averaged reference with 14 channels. EEG for pediatric and adults subjects utilized standard 10–20 system with averaged reference montage with total 19 channels. For calculation of EEG spectra we calculate the power spectral density (PSD) using Welch's method in 100 frequency bins between 0 to 50 Hz, which is less than half the sampling rate for all EEGs. This pre-processing was performed with the python MNE package[111]. To constrain dimensionality, we utilize the global mean across all channels for both observed and simulated spectra. EEG measurements were taken from distinct samples from each subject.

For SGM spectral realizations used in the main analysis, we utilized the adult template structural connectome obtained from the MGH-USC Human Connectome Project (HCP) database[40]. We compared SGM realizations derived from the adult HCP connectome to those derived from a template neonatal connectome. The template neonate connectome was derived from 20 full-term neonates from the developing HCP (dHCP)[112].

## SBI of SGM parameters

A challenge central to neural modeling is the statistical inference of simulation model parameters, with respect to observable data, which in Bayesian inference, can arise due to computational intractability of the posterior distribution for high dimensional systems or analytical intractability of the likelihood function[113]. In addition, estimation of brain model parameters of electrophysiological maturation from observed EEG constitutes a classic inverse problem characterized by non-uniqueness: multiple biologically plausible brain model specifications that generate similar EEG maturation trajectories. In this regard, the Bayesian approach provides a suitable framework for incorporating biophysically plausible priors as constraints and subsequent inference of SGM parameter posterior distributions[114]. Recent advances in SBI leveraging deep learning advances to approximate the posterior distributions for respective simulation parameters have shown promise in application to single-neuron mechanistic models[39], and more broadly have seen increasing utility in diverse scientific fields including particle physics and astrophysics[113]. The potential of SBI to model high-dimensional whole-brain dynamics has also recently been leveraged in modeling epilepsy and in healthy brain aging[106,115].

Modeling neural data is hindered by the inaccessibility of the ground truth set of biophysical parameters across the microscopic to macroscopic continuum that determine the human brain's structural and functional state. Furthermore, even if these parameters were available, the likelihood function becomes computationally intractable in the face of high-dimensional complexity characteristic of neural data. Moreover, traditional MCMC methods used for Bayesian inference, while powerful are computationally inefficient, particularly when applied to high-dimensional neural modeling and may have convergence issues[116]. In this context, recent developments in probabilistic models in machine learning (ML) have led to methodological advances in SBI, including deep-learning based compact representation of high-dimensional data, active learning methods to simulate at parameters $\theta$ that have higher likelihood of increasing knowledge the most, and amortization of probability density estimation[113]. In our approach, we evaluated three recent SBI techniques: Neural Ratio Estimation (NRE)[42], Neural Posterior Estimation (NPE)[43], and Truncated Sequential Neural Posterior Estimation (TSNPE)[44]. In NRE, a neural network classifier is trained to approximate the likelihood-to-marginal ratio[42]. In NPE, which is also known as automatic posterior transformation (APT) and sequential NPE (SNPE) when utilized in multi-round inference contexts, the generative model is run sequentially across different sets of parameters generating model behavior on which a deep neural network can be trained to output a posterior distribution over the input model parameters[43]. This approach, which circumvents direct computation of the likelihood function, has recently been effectively applied in the domain of functional to structural brain modeling in the context of healthy brain aging[106]. We also compare standard NPE to the recently introduced

truncated NPE (TSNPE), which more robustly handles posterior estimation near the margins of the specified prior range compared to NPE[44]. While we focus on recent SBI methods here, we acknowledge that there are alternative robust methods for parameter estimation. A prominent Bayesian inference approach is the Hamiltonian Monte Carlo (HMC) algorithm, which offers superior convergence and robustness in handling high-dimensional spaces and models with highly correlated parameters relative to traditional MCMC methods[117]. In contrast to SBI, which enables approximate Bayesian inference even when the likelihood function is unavailable, HMC requires availability of the likelihood function, as it operates within a likelihood-based framework of Bayesian inference. Furthermore, model differentiability is optional with SBI whereas HMC requires it. HMC employs gradient calculations of the log posterior with respect to the model's parameters to compute the Hamiltonian dynamics, facilitating efficient exploration of the posterior distribution. Additionally, HMC may be utilized to sample the posterior constructed with differentiable, likelihood-based SBI techniques such as NLE. This approach facilitates robust sampling and efficiently scales to high-dimensional parameter spaces[118]. A compelling demonstration of these gradient-based HMC methods in combination was demonstrated by Hashemi et al. for the inversion of a differentiable whole-brain model to model subject-specific epileptogenicity[119]. The SGM is differentiable; however, caution and further investigation are needed to evaluate the stability of eigendecomposition gradients due to parameter degeneracy and the use of non-Hermitian matrices in the SGM.

Given the observed EEG spectra $y$, the respective SGM parameter posterior distribution $p(\theta \mid y)$ was determined by NRE, NPE, or TSNPE utilizing the *sbi* Python package[43,120]. In this approach, SGM is evaluated sequentially across specified parameter value ranges generating SGM spectral output realizations on which the neural density estimator is trained to approximate the posterior distribution over the input model parameters (Fig. 1b). Prior knowledge is parameterized in the SGM as the range of admissible values for $\theta$ based on biophysically plausible values following Raj et al.[38]. SGM parameters and their respective, physiologically-informed bounds used for SBI are listed in Table 1. We evaluated varying simulation budget sizes up to 1E6 simulations to train the neural density estimator for NRE and NPE. For TSNPE, we evaluated varying budget sizes up to 2000 simulations at two or three rounds of sequential inference with proposal truncation. We utilized summary statistics of SGM power spectral density (PSD) output, the first periodic and aperiodic components (Supplementary Fig. 1c) which succinctly parameterizes physiologically and developmentally pertinent EEG spectral features[18,20], and binned PSD as utilized by Rodrigues et al. for neural mass model (NMM) inversion with Hierarchical Neural Posterior Estimation[121]. The resulting inferred posterior distributions $p(\theta \mid y)$ contain high values for parameters $\theta$ that are consistent with the empirical data $y$ and are asymptotic to zero for $\theta$ inconsistent with $y$. 10000 samples (i.e., consistent SGM parameter sets) per subject were drawn from the posterior $p(\theta \mid y)$ to determine the multivariate posterior distribution. The distributions of $p(\theta \mid y)$ over $y$ observed across different ages were used to evaluate longitudinal changes of $\theta$ and for incorporation into subsequent regression models.

## Bayesian sensitivity analysis

We apply Bayesian identifiability and sensitivity analyses to assess pathology in the inference approach and to characterize the performance of the inference procedure[122]. We evaluate the posterior shrinkage[122], defined as $s = 1 - \frac{\sigma_{\text{post}}^2}{\sigma_{\text{prior}}^2}$. If the posterior closely resembles the prior without significant shrinkage, it indicates structural non-identifiability, as the data does not inform our understanding of these parameters. Conversely, when the posterior is influenced by the data (evidenced by shrinkage) but demonstrates a strong statistical interdependence among parameters, this indicates either structural or practical non-identifiability, characterized by an inability to uniquely determine individual parameters due to their interrelated nature in the model[123]. In addition, the posterior $z$-score may be used to assess how accurately the posterior recovers ground truth parameters. We evaluate the

posterior $z$-score[122], defined as: $\mathbf{z} = |\frac{\mu_{post} - \bar{\theta}}{\sigma_{post}}|$ where $\mu_{post}$ is the posterior mean, $\sigma_{post}$ is the posterior standard deviation, and $z$ quantifies how accurately the posterior mean approximates the true parameter.

## Simulation-based Calibration of SBI-SGM

Simulation-Based Calibration (SBC) is used to validate the precision of uncertainties within the Bayesian framework by assessing whether the variance in the posterior distribution, which represents uncertainty in model parameters, is accurately calibrated[124]. SBC involves generating multiple observational datasets from a range of parameters drawn from the prior distribution, then for each dataset, a posterior distribution is computed using SBI. The subsequent posterior calibration is deemed accurate if the ranks of the original parameters, when evaluated within their corresponding posterior distributions, collectively exhibit a uniform distribution. The uniformity of the resulting SBC ranks is then visually assessed by comparing their empirical cumulative density function to that of an ideal uniform distribution, ensuring the reliability of the model's uncertainty estimations. This uniformity of the normalized rank statistics aligns with a necessary, though not sufficient, condition for the estimated posterior to be accurate. We utilize visualization of ranks cumulative density function in comparison to the uniform distribution, and also utilize the one-sample Kolmogorov-Smirnov test to check whether the ranks drawn from the estimated posterior follow the normal distribution. In addition, we utilize Classifier 2-Sample Test (C2ST) to compare the estimated posterior to the prior distributions[125]. We utilized the python *sbi* package implementation of C2ST, which utilizes a binary MLP classifier to distinguish samples from the estimated posterior $\theta \sim \mathbf{q}(\theta|\mathbf{x})$ and the reference posterior $\theta \sim \mathbf{p}(\theta|\mathbf{x})$. The subsequent test accuracy of the classifier ranges from 0.5, indicating that if accuracy is not better than chance then the ensembles are drawn from the same distribution, to 1 where the distributions are exactly distinguishable.

## Evaluating association of temporal evolution of SGM parameters with age

To evaluate the dynamics of SGM parameters over time during development, we tested for the correlation of SGM parameters with age. For each subject, the mean value of the inferred SGM parameter posterior distribution was selected. We then used Pearson Product-Moment Correlation to evaluate the linear association between age (log years) and SGM parameters $\alpha$, $S$, $G_{EI}$, and $G_{II}$. Next, we sought to validate our modeling approach by evaluating the ability of SGM parameters obtained automatically via SBI to predict age, PDR, and the aperiodic exponent. We regressed predicted vs observed values, on the x and y axes, respectively, following Piniero et al. for prediction of age, PDR, and aperiodic exponent[126]. The PDR and aperiodic exponent were automatically detected using the Fitting Oscillations & One Over F (FOOOF) Python package as demonstrated in Supplementary Fig. 1a[18]. We utilized a polynomial regression model (degree = 2) with k-fold cross-validation (k = 10). This model selection was motivated by the presence of nonlinear changes in different brain parameters over time. Such nonlinear dynamics in brain development have been recently modeled using polynomial approaches[3]. Prior functional connectivity studies have also tended to model nonlinearity with polynomial regression[127]. SGM parameters used in the regression model included those found to correlate with age. During each cross-validation fold, the polynomial regression model is fit to the training set containing these SGM parameters, and their mean square error (MSE) in age prediction and coefficient of determination is calculated on the held-out set.

## Reporting summary

Further information on research design is available in the Nature Portfolio Reporting Summary linked to this article.

## Data availability

EEG data were obtained from publicly available neonatal and infant EEG datasets with an age range of one month to one year of age[109] and publicly available database of EEGs containing ages 5 to 30 years of age[110]. The power spectra derived from the publicly available EEG samples used for SBI are available from the corresponding author upon reasonable request. EEG data for subjects between one day and five years were obtained at the University of California San Francisco (UCSF) and raw patient-related data are not available due to data privacy laws. Pre-processed UCSF EEG data are available under restricted access due to ethical and privacy reasons. UCSF EEG data can be requested by contacting the corresponding author, and data sharing is conditional to the establishment of a specific data-sharing agreement between the applicant's institution and UCSF.

## Code availability

Code for the spectral graph model (SGM) can be found on the spectrome package GitHub page (https://github.com/Raj-Lab-UCSF/spectrome). Code for incorporation of the SGM with SBI using the *sbi* Python package with examples are available on the project's GitHub page (https://github.com/dbernardo05/sbi-spectrome).

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

## Acknowledgements

Pew-Thian Yap was supported in part by the National Institutes of Health (NIH) under grants R01EB008374, R01MH133836, and R01MH125479. Ashish Raj (AR) and Parul Verma were supported by NIH grant R01AG072753 (AR). Srikantan Nagarajan (SSN) was supported by NIH grants RF1NS100440 (SSN), R01DC017091 (SSN), and P50DC019900 (SSN). Figure one was created with BioRender.com.

## Author contributions

D.B. performed the data analyses, drafted the manuscript, and was involved in study conceptualization and design. X.X., P.V., and J.K. contributed to drafting the manuscript. V.L., A.L.N., and H.C.G. contributed to data collection. P.T.Y. and Y.W. contributed to data collection and analysis. S.S.N. was involved in study design and revised the manuscript. A.R. was involved in study conceptualization and study design. All authors reviewed and revised the manuscript.

## Competing interests

The authors declare no competing interests.
