## [Peer Review File · Communications Physics]

Reviewers' comments:

Reviewer #1 (Remarks to the Author):

This article tackles a fascinating topic by an excellent chosen approach the so-called Simulation-based inference (SBI). Bernardo et al, have used SBI to estimate age-varying

parameter posterior probability distributions from EEG spectral changes. Although methods used here such as SBI and Bayesian estimation from spectral power have been previously developed by other teams (SBI package from Mackey lab and fitting spectral EEG using spectral DCM), the application here is robust and very interesting but at the end with a general conclusion; spectral maturation of brain activity is an emergent phenomenon guided by the age-dependent tuning of localized neuronal dynamics modulated. I see this a somewhat circular statement as there are changes in observation and model parameters varied by such changes. This may limit the impact of the work such that only the model fitting accurately captures the EEG spectral features and interpretation of inferred model parameters is left to the reader. More critically, a single structural connectome template was used across all subjects. How the causal changes in parameters with age can be supported by using a common connectome? Thus, the approach is not personalized. Nevertheless, I read the paper with interest particularly for the inference on longitudinal EEG oscillatory changes. Some issues need however to be addressed before re-considering its appropriateness for publication, in particular, first the validation on synthetic data to show the recovery of ground-truth parameters.

Major

Table 1: I don't see the bounds for the time constant parameters. Critically, why the excitatory/inhibitory gains are truncated at 5 but not zeros? SBI may get stuck at the bounds? The main reason to ask is that SBI may get the tendency to estimate the parameters as zeros if there is no sensitivity to them (non-identifiability)? For instance, in Fig 2, G_{II} is not sensitive to delta and alpha peaks. How the biological constrains on parameters are satisfied, e.g., response functions for the excitatory synapses exhibit a longer characteristic rise and decay times than the inhibitory synapses (see Hashemi et al, Optimal model parameter estimation from EEG power spectrum features observed during general anesthesia, *Neuroinformatics* 16.2 (2018): 231-251.) Moreover, the range for the conduction velocity seems very low compared to other estimates, e.g. see Lemarechal et al *Brain* 2022.

Fig2. Please clarify the prior for each parameter, the running time and number of simulations used for SBI. I am really curious to see an example on synthetic data to see how SBI recovers the ground-truth values. In particular, the level of posterior shrinkages to investigate the non-identifiability of parameters. Please note that using SBI, the posterior is always fall in the ranges provided using prior, and critically the level of posterior shrinkages determine the reliability of the estimation. A demonstration with having ground-truth value will be essential for validation of convergence and very helpful in the context of identifiability analysis, particularly as validation is mentioned in line 58.

Fig 4. The Bayesian inference with SBI is performed to capture the uncertainty on the estimation, however, this has not been shown at the subject level. The posterior distribution needs to be illustrated. The importance of identifiability analysis manifest here as if a model parameter is not sensitivity to EEG features, then it becomes controversial to make hypothesis on parameter changes in terms of aging. However, this is interesting that long-range coupling strength increases with ages as demonstrated in Lavanga, Mario, et al. "The virtual aging brain: a model-driven explanation for cognitive decline in older subjects." bioRxiv (2022).

Fig 5. Here, it is very vague how the subject's age is inferred from EEG features. Cross-validation is performed after SBI, or for each hold-out section, the training is repeated. Moreover, what is the assumption behind considering only polynomial regression?

Line 213: I was reading the paper with interest that the work provided the personalized connectivity matrix in model, then I see it here, it is a single structural connectome template was used across all subjects? So, how the conclusion about the parameter changes with age can be supported? The connectome changes dramatically with ages.

What does biologically indicate the alpha parameter progression? It seems that it is mainly this parameter that reflect the alpha power in EEG, however as pointed out above, the same connectome has been used for all the subjects?! Then how the model is personalized? Then critically, why the model showed the need for larger coupling?

Minor

Introduction: Maturation is degeneration or deterioration in brain connectivity?

Line 20; please provide citation for SBI such as Cranmer et al PNAS 2020, Gonçalves et al eLife 2020.

Why SBI have been used in this study? Is it because the likelihood is intractable to use Monte Carlo sampling? If the tracking in parameter changes is the interest why not to use Kalman filtering?

It is worth elaborating on spectral graph model rather than neural-field or neural mass modelling merged with brain connectome such as virtual brain modelling approach (e.g., Hashemi, Meysam, et al. "The Bayesian Virtual Epileptic Patient: A probabilistic framework designed to infer the spatial map of epileptogenicity in a personalized large-scale brain model of epilepsy spread." *NeuroImage* 217 (2020): 116839.

Also, in the following recent study, a causal link between structural connectivity and brain function for healthy aging and SBI is introduced. However, the work is related to fMRI resting state rather than EEG and it is not also longitudinal investigation: Lavanga, Mario, et al. "The virtual aging brain: a model-driven explanation for cognitive decline in older subjects." *bioRxiv* (2022).

Section 2.1. How many subjects? How many channels/sensors are located for EEG? As I don't see any lead-field (projection) matrix, I assume that the results are at source level. The estimation at sensor will be much more challenging!

Line 78. SGM is suitable for capturing changes in only aperiodic activity?

Line 89: I do not see Figure 2G.

Line 142. In SBI, particularly SNPE, there is no true posterior! Otherwise, the KL divergence (or the free energy) can be reported? Rather, the motivation is intractability of likelihood function, also to avoid the sensitivity to the features in classical ABC approach. See Cranmer et al PNAS 2020.

Line 156. This is in agreement with Lavanga et al to explain the brain fluidity changed as a function of age and global neuromodulation due to an increase in modulation between structural connectivity and brain function as a result of interhemispheric degradation of structural connectivity?

Line 158. Indeed, none of the cited works (refs 42-46) are whole-brain modeling, rather these are constrained as coupled a few neural mass models (coupled cortical microcolumn or thalamocortical system but not at the whole-brain scale).

Line 208: what is the justification for using linear filters for the signal transmission? Nonlinear models have been shown to be crucial for modelling the brain at rest or pathologies (Sanz Leon et al 2015) such as epilepsy (Hashemi et al Neuroimage 2020). A recent normalization of the connectome (Petkoski & Jirsa 2022 Netw Neurosci), showed that time-delays can be responsible for the appearance of PDR.

In the case of EEG, using the linear models, it is time delay that make it feasible to get multiple frequency bands such as alpha and beta. How one can get simultaneous EEG peaks with linear models and no time delay?

Line 216 and 221; with a coupled linear delay differential equation, one can get both aperiodic and periodic components and this not a specific feature of the model used in this study and can be seen also in other neural mass modelling (even not at the whole brain level, rather thalamo-cortical system such as Hashemi et al Neuroinformatics 2018).

Line 307: Please see Lavanga, Mario, et al. "The virtual aging brain: a model-driven explanation for cognitive decline in older subjects." bioRxiv (2022).

Typo line 324, space after).

Line 326: I see it for 16000 simulations in appendix.

Reviewer #2 (Remarks to the Author):

Summary:

A major goal of computational cognitive and neuroscience is to produce models with few parameters which can account for significant aspects of behavioral, neural or physiological data, making otherwise hard to discern patterns interpretable and hence understandable. Bernardo et. al. contribute to this general agenda a computational account of developmental aspects of neural activity patterns. The authors perform simulation-based inference on EEG spectral features with the Spectral Graph Model (SGM), and identify parameters of this model that consistently vary with age. They demonstrate that varying specific SGM parameters can recapitulate shifts in the Posterior Dominant Rhythm (PDR) and spectral slopes in accordance with maturation patterns observed in empirical studies, suggesting a mechanistic explanation for the spectral maturation of neural activity.

Main review:

Motivated by an analysis of the indeterminacy of the inverse problem (from data to parameters of the SGM) the authors opt for approximate bayesian inference to fit the SGM to subject data. We consider the findings as surprising, significant and an important contribution to the developmental part of the computational neuroscience literature. A computational account, which lets us, through the lens of a generative model and specifically interpretable parameter variations, appreciate important aspects of brain maturation can form significant connective tissue amongst a plethora of empirical findings across age groups.

In the following we however would like to point out some, mostly methodological, critiques which we would like to see addressed to bolster the credibility of the stated results. We address the major points below and then follow with a list of minor comments.

SBI analysis:

1. In general the sections concerning SBI analysis could benefit from clearer motivation. What about the inference problem necessitates simulation based inference to begin with? The main SGM paper cited (Raj. et. al. 2020) does not emphasize simulation based inference. Raj et. al. (2020) moreover emphasize that the SGM is deterministic with a closed form solution in the Fourier domain. However, if the model is deterministic and uniform priors are used (which we understood the authors do use), a posterior should in fact be uniform over the region of admissible parameters. Posteriors illustrated in the paper do never seem uniform (and therefore should be rather bad approximations of true posteriors), however it is reasoned that the simulation budgets were sufficient. Can you clarify this? As also suggested below in the UMAP critique, parameter recovery studies could help the reader understand the quality of the posteriors. A useful technique to test your inference pipeline is ‘simulation based calibration’ (Talts et. al., 2018). Lastly, given the above, you may also resort to the MAP estimation technique of Raj. et. al. (2020), as a comparison.

2. As per Figure 3, the authors use the peaks of marginal posterior distributions as the estimates of parameters on which later regression models are based. The vector of peaks of posterior marginals may in fact not represent the MAP vector. This choice should be motivated more strongly. Moreover, information about the full posteriors is essentially never used (for example, why not run the regression analysis on the basis of posterior mean estimates?).

3. Figure S2 shows examples of posteriors under increasing simulation budgets. If the motivation of this figure is to show how posteriors stabilize after a simulation threshold is crossed, it fails to convey this message. In fact the posterior distribution for 8000 simulations looks quite different from the posterior distribution which used 16000 simulations. To illustrate its (assumed) point, the figure should show graphs based on simulation budgets which cross the point at which no further change in the posterior is elicited from increases in the simulation budget.

4. The authors motivate their choice of the data summary statistic by stating that the first periodic and aperiodic components of the SGM spectral output “succinctly parameterizes physiologically and developmentally pertinent EEG spectral features”. This is reasonable, but also heuristic. Given that all their analyses and conclusions rely heavily on this choice, it seems important to characterize how sensitive the inference procedure is with respect to this choice.

Regression analysis:

1. The regression analysis reported in Figure 4, opens some questions regarding fulfillment of basic assumptions of the regression models. For Figure 4a), 4b) and 4c) one can argue that the basis of the increase/decrease in parameter values across ages, might not stem from an actual increase in

mean, but rather an increase in variance of dependent variables across age. The estimates of these variances are in fact available since the chosen parameter values are derived from posterior distributions.

2. The parameter values used for this analysis have to be considered in light of the critique of the SBI analysis. The regression was based on the maxima of the approximate marginal posteriors. Instead, the parameter values corresponding to the actual MAP estimates could have been used. Another standard approach is to consider the posterior mean estimates. It would be essential to see whether the findings are robust to these variations here.

3. In Figure 5 (b,c) the authors illustrate how a polynomial regression on relevant fitted model parameters fares in predicting the PDR and aperiodic exponents of observed data. To establish expectations, it will be useful to include a prediction of age and/or PDR and aperiodic exponents directly from the summary statistics. How good / bad is the level of inference afforded via the SGM parameters in comparison?

4. A polynomial regression of model parameters to predict main features (PDR, aperiodic exponent) of the EEG spectra from SGM parameters was only marginally successful. Is this because of the choice of summary statistic on which the posterior inference was based? Again it is unclear what level of success should be expected.

UMAP analysis:

The UMAP analysis is used to argue that the SGM may exhibit indeterminacy issues. This argument seems circular. Instead, some parameter recovery studies (perhaps from synthetic data simulated using SGM) could be performed which illustrate e.g. strong parameter trade-offs if data is specified via summary statistics as chosen in the paper. These parameter studies can also inform some of the interpretations regarding roles of respective parameters implicit in the (forward simulation) analysis presented in Figure 2. Figure 2 can only show univariate claims (varying one, hold other parameters constant) which do not necessarily elucidate more complicated parameter tradeoffs that could emerge when dealing with the inverse problem (parameters might compensate in complex fashion to produce given patterns).

Minor comments:

73-75, discussion of 2B refers to 2C and vice versa

81-82, emphasis should be on “variation in parameters” of the SGM that can account for developmental trajectories.

95-96, the neural density estimators (NDEs) are trained to provide approximate posteriors. Granted if you use uniform priors in principle NDEs should provide you with a uniform posterior over admissible parameters but only if your model is noise-free and suffers from any parameter non-identifiability (degeneracy). The wording here is misleading.

141-143 The accuracy of the posterior does not measure model performance, the posterior predictive may. If “model” here indicates the NDE (so the approximate posterior itself), this should be specific to avoid ambiguity.

193 Two models being special cases of each other seems logically implausible.

206-208 If linear models actually perform better than non-linear models, then no extra motivation is needed to use them.

308-310 The fact that multiple parameter combinations may be able to generate similar data is not in and of itself what makes direct calculation of the posterior problematic. If you do not have a likelihood function but can only simulate from the model, you cannot “calculate the posterior” as in you cannot evaluate it and need to resort to likelihood free methods (SBI for example). If you do have a likelihood you may want to use MCMC to sample from the posterior, which is complicated by multimodality and can be complicated by high correlations (two facets of indeterminacy), but you can evaluate the posterior or an object proportional to it.

590 (Pg. 28) Axis titles are too small. Figure S1 a), age in years not days.

RESPONSE TO REVIEWERS

We appreciate the Reviewer' commentary and insightful feedback. In summary, we have added simulation-based calibration (SBC), parameter recovery analyses, and comparisons between SNPE and the more recent truncated SNPE (TSNPE). In addition, to address concerns that a single structural connectome was used for all subjects, we compared SGM realizations derived from adult and neonatal averaged connectomes. We have addressed the concerns raised below point by point.

Reviewer #1 (Remarks to the Author):

This article tackles a fascinating topic by an excellent chosen approach the so-called Simulation-based inference (SBI). Bernardo et al, have used SBI to estimate age-varying parameter posterior probability distributions from EEG spectral changes. Although methods used here such as SBI and Bayesian estimation from spectral power have been previously developed by other teams (SBI package from Mackey lab and fitting spectral EEG using spectral DCM), the application here is robust and very interesting but at the end with a general conclusion; spectral maturation of brain activity is an emergent phenomenon guided by the age-dependent tuning of localized neuronal dynamics modulated. I see this a somewhat circular statement as there are changes in observation and model parameters varied by such changes. This may limit the impact of the work such that only the model fitting accurately captures the EEG spectral features and interpretation of inferred model parameters is left to the reader. More critically, a single structural connectome template was used across all subjects. How the causal changes in parameters with age can be supported by using a common connectome? Thus, the approach is not personalized. Nevertheless, I read the paper with interest particularly for the inference on longitudinal EEG oscillatory changes. Some issues need however to be addressed before re-considering its appropriateness for publication, in particular, first the validation on synthetic data to show the recovery of ground-truth parameters.

We deeply appreciate the overall feedback of the Reviewer. We have added a comparison between the neonatal and adult averaged connectomes which indicate that postnatal structural connectome development exerts a minor effect size relative to the changes that arise from age dependent SGM parameter changes. This observation indicates that postnatal EEG maturational change may derive primarily from functional, rather than structural, network changes. Taking note of this finding, in order to avoid circularity, we have altered our conclusion as stated in the Abstract and Discussion to

indicate that: the observed developmental trajectory of brain spectra may be attributed principally from functional, as opposed to structural, adaptations.

Major

Table 1: I don't see the bounds for the time constant parameters. Critically, why the excitatory/inhibitory gains are truncated at 5 but not zeros? SBI may get stuck at the bounds?

We have added the bounds used in the table. We have set bounds ranged based on prior studies as well as our work with the SGM which established bounds that recapitulated physiological EEG/MEG spectra.

The main reason to ask is that SBI may get the tendency to estimate the parameters as zeros if there is no sensitivity to them (non-identifiability)? For instance, in Fig 2, G_{II} is not sensitive to delta and alpha peaks. How the biological constraints on parameters are satisfied, e.g., response functions for the excitatory synapses exhibit a longer characteristic rise and decay times than the inhibitory synapses (see Hashemi et al, Optimal model parameter estimation from EEG power spectrum features observed during general anesthesia, Neuroinformatics 16.2 (2018): 231-251.)

Zeros were not chosen as the lower bounds as we selected the bounds on biophysically compatible values. Indeed, we did find that the original analysis with SNPE-C did tend towards systematic under-estimation of the posterior mean and hence provided poorly calibrated cumulative density functions during simulation based calibration. We subsequently utilized truncated SNPE which appears to alleviate SBI issues close to parameter bounds.

In response to the reviewer's example of differential response functions for excitatory and inhibitory synapses, which are enforced during spectral fitting in Hashemi et al., within the proposed SBI-SGM framework, we do not directly enforce biological constraints with the aim of a parsimonious model of the spectral brain activity. We enter this into the discussion in comparison to Hashemi et al. as a limitation of our study and potential future direction.

Moreover, the range for the conduction velocity seems very low compared to other estimates, e.g. see Lemarechal et al Brain 2022.

This was a typo and the max bound of conduction velocity initially used was 5 m/s. 5 m/s is within the range of the reported median values and median absolute deviations of axonal conduction velocity reported by Lemaréchal et al, 3.1 ± 2.0 m/s and 3.5 ± 1.8 for the <15 yo and >15 yo age groups, respectively.

For the re-analysis, we have increased the upper bounds to 15 to include the broader range of axonal conduction velocity reported by Lemaréchal et al.

Fig2. Please clarify the prior for each parameter, the running time and number of simulations used for SBI. I am really curious to see an example on synthetic data to see how SBI recovers the ground-truth values. In particular, the level of posterior shrinkages to investigate the non-identifiability of parameters. Please note that using SBI, the posterior is always fall in the ranges provided using prior, and critically the level of posterior shrinkages determine the reliability of the estimation. A demonstration with having ground-truth value will be essential for validation of convergence and very helpful in the context of identifiability analysis, particularly as validation is mentioned in line 58.

Following Betancourt et al. 2018, we computed posterior shrinkage as well as posterior z-scores which evaluate deviation of estimated means from true means on synthetic data. We have included this analysis in Supplementary Figures 5 and 6. We have also included the running time for TSNPE at different simulation budget sizes in Supplementary Table 1.

Fig 4. The Bayesian inference with SBI is performed to capture the uncertainty on the estimation, however, this has not been shown at the subject level. The posterior distribution needs to be illustrated. The importance of identifiability analysis manifest here as if a model parameter is not sensitivity to EEG features, then it becomes controversial to make hypothesis on parameter changes in terms of aging. However, this is interesting that long-range coupling strength increases with ages as demonstrated in Lavanga, Mario, et al. "The virtual aging brain: a model-driven explanation for cognitive decline in older subjects." bioRxiv (2022).

We have illustrated the posterior distribution across all parameters for all subjects at an individual subject level in Supplemental Figure 9A and 9B.

Fig 5. Here, it is very vague how the subject's age is inferred from EEG features. Cross-validation is performed after SBI, or for each hold-out section, the training is repeated. Moreover, what is the assumption behind considering only polynomial regression?

We have clarified the details regarding polynomial regression methodology in Methods section 4.7. SBI was performed once prior to cross-validation. During cross-validation fold, each polynomial regression model is fit to the training set then tested on the hold-out set.

We have also added the rationale for polynomial regression model selection to the Methods section 4.7: The choice of polynomial regression was motivated by the

presence of non-linear changes of different brain parameters over time. Such non-linear dynamics in brain development have been recently modeled using polynomial approaches (Bethlehem, et al. Brain charts for the human lifespan. Nature 2022). Additionally, prior functional connectivity studies have tended to model non-linearity with polynomial (Sanders, Ashley FP, et al. Age-related differences in resting-state functional connectivity from childhood to adolescence. Cerebral Cortex 2023).

Line 213: I was reading the paper with interest that the work provided the personalized connectivity matrix in model, then I see it here, it is a single structural connectome template was used across all subjects? So, how the conclusion about the parameter changes with age can be supported? The connectome changes dramatically with ages.

This points out a limitation of our approach, given that the structural connectome can change with age. We decided to employ a single structural connectome for two primary reasons:

- 1) We fully agree that age-specific structural connectomes ideally should be used, however, at present, there is a paucity of diffusion MRI data for the 1 month to 5 year-old age-range which precludes this.
- 2) Recent work has illustrated the respective changes of the structural and functional connectomes in aging and suggest that the structural connectome may be relatively stable after birth. The core components of the structural connectome develop in utero and there is little reorganization postnatally (Ciarrusta, Judit, et al. "The developing brain structural and functional connectome fingerprint." *Developmental Cognitive Neuroscience* 2022). Moreover, the functional connectome appears to be present in early infancy and demonstrates stability at an early age (Hu, Dan, et al. "Existence of Functional Connectome Fingerprint during Infancy and Its Stability over Months." *Journal of Neuroscience* 2022). Thus, despite the inherent limitations in utilizing a single structural connectome, our findings align with recent findings on the stability of a core structural network and provides insight into how functional changes arise upon a consistent structural framework. In other words, postnatal EEG maturation may primarily derive from functional changes rather than structural connectome changes.

In addition, to evaluate the effect that utilization of a single structural connectome for all subjects may have on SGM realizations, we compared SGM realizations derived from adult and neonatal population-averaged connectomes. These findings (Supplementary Figure 3) indicate that postnatal structural connectome development exerts a minor effect size relative to the changes that arise from age dependent SGM parameter changes. This observation indicates that postnatal EEG maturational change may

derive primarily from functional, rather than structural, network changes. Nevertheless, we view the unavailability of age-dependent as a limitation of our study and have added this to the limitation section of our Discussion.

What does biologically indicate the alpha parameter progression?

Alpha parameter progression physiologically reflects that at a local level, there is an increase in the weighting of afferent long-range cortico-cortical connections. Biologically, this is mediated by increased long-range excitatory and GABAergic coupling. (Urrutia-Piñones, Jocelyn, et al. Long-Range GABAergic Projections of Cortical Origin in Brain Function. *Frontiers in Systems Neuroscience* 2022; Sydnor et al. Neurodevelopment of the association cortices: Patterns, mechanisms, and implications for psychopathology. *Neuron* 2021.)

We have added this citation to Discussion Section 3.2.

It seems that it is mainly this parameter that reflect the alpha power in EEG, however as pointed out above, the same connectome has been used for all the subjects?! Then how the model is personalized? Then critically, why the model showed the need for larger coupling?

In our reply above, we cite recent literature that suggests that the core of the brain's structural network is primarily developed in utero and remarkably stable postnatally during early brain development. Notably, in the SGM, the gain and alpha parameters reflect neural dynamics that occur on a timescale of seconds (Verma et al. *Network Neuroscience* 2023). The result that a gradual increase in the alpha parameter over time recapitulates the monotonic increase in posterior dominant rhythm (PDR) seen during infancy to young adulthood, suggests that the observed developmental increase in PDR is in part driven by a gradual increase in resting tone of long-range functional coupling within the connectome. Biologically this may be mediated by increased long-range excitatory and GABAergic coupling. (Urrutia-Piñones, Jocelyn, et al. Long-Range GABAergic Projections of Cortical Origin in Brain Function. *Frontiers in Systems Neuroscience* 2022; Sydnor et al. Neurodevelopment of the association cortices: Patterns, mechanisms, and implications for psychopathology. *Neuron* 2021.)

We have modified our Discussion Section 3.4 to clarify this point.

As stated above, we have performed additional analysis and show that differences in the structural connectome between neonates and adults does not appear to lead to significant change in spectral power distribution. In Supplementary Figure 3, we compare spectral power SGM realizations derived from an average neonatal structural connectome from the dHCP dataset, compared to the corresponding adult connectome.

Minor

Introduction: Maturation is degeneration or deterioration in brain connectivity?

In the developmental context studied here, between the neonate to young adult (<30 yo), at which point canonical functional networks are considered fully developed, we view maturation in this context as generation of increased functional coupling upon the structural connectome.

Line 20; please provide citation for SBI such as Cranmer et al PNAS 2020, Gonçalves et al eLife 2020.

We have added this.

Why SBI have been used in this study? Is it because the likelihood is intractable to use Monte Carlo sampling? If the tracking in parameter changes is the interest why not to use Kalman filtering?

- We have previously utilized Monte Carlo sampling in our prior work and SBI offers a significantly more computationally efficient approach.
- A Kalman filter may be applied to predict individual SGM parameter changes over time, however, in our use case we were interested in population-based trajectories of SGM parameters. While it appears there are recent population-based Kalman filter approaches to model infectious disease, these are not yet developed for population-based studies of brain development.

It is worth elaborating on spectral graph model rather than neural-field or neural mass modelling merged with brain connectome such as virtual brain modelling approach (e.g., Hashemi, Meysam, et al. "The Bayesian Virtual Epileptic Patient: A probabilistic framework designed to infer the spatial map of epileptogenicity in a personalized large-scale brain model of epilepsy spread." NeuroImage 217 (2020): 116839. Also, in the following recent study, a causal link between structural connectivity and brain function for healthy aging and SBI is introduced. However, the work is related to fMRI resting state rather than EEG and it is not also longitudinal investigation: Lavanga, Mario, et al. "The virtual aging brain: a model-driven explanation for cognitive decline in older subjects." bioRxiv (2022).

The spectral graph model (SGM) diverges from methods that graft the connectome to neural-field or neural mass models by utilizing structural connectome eigenmodes. Specifically, SGM leverages these connectome eigenmodes in conjunction with neural-field model to capture the spatial distribution of neural activity. This approach is substantiated by recent research demonstrating that these eigenmodes can effectively reconstruct the spatial patterns of canonical functional networks (Atasoy, S., et al., 2016; Preti, M., et al., 2019). Moreover, our application of SGM has shown that when a neural-field model is propagated upon these connectome eigenmodes, it successfully reproduces the spatial spectral distributions observed in MEG (Raj A, et al 2020).

Section 2.1. How many subjects? How many channels/sensors are located for EEG? As I don't see any lead-field (projection) matrix, I assume that the results are at source level. The estimation at sensor will be much more challenging!

A total of 234 subjects were included, with each subject having the standard 10-20 EEG montage.

Line 78. SGM is suitable for capturing changes in only aperiodic activity?

The SGM is able to capture frequency distributions across the entire spectrum, including periodic activities in the alpha range.

Line 89: I do not see Figure 2G.

We have corrected this.

Line 142. In SBI, particularly SNPE, there is no true posterior! Otherwise, the KL divergence (or the free energy) can be reported? Rather, the motivation is intractability of likelihood function, also to avoid the sensitivity to the features in classical ABC approach. See Cranmer et al PNAS 2020.

We appreciate this feedback and have corrected this paragraph to emphasize intractability of the likelihood function in the motivation of utilizing SBI.

Line 156. This is in agreement with Lavanga et al to explain the brain fluidity changed as a function of age and global neuromodulation due to an increase in modulation between structural connectivity and brain function as a result of interhemispheric degradation of structural connectivity?

We appreciate the reviewer drawing a parallel to the work of Lavanga et al. In contrast to Lavanga et al. which included a population with age range 18-87, our study focused on neurodevelopmental network changes from the neonatal period to young adulthood. Whereas between the young adult age to geriatric age, degradation of structural connectivity is expected, within the age range of our study, maturation of structural connectivity is expected. We have commented on our work's agreement with Lavanga et al and added this recent reference.

Line 158. Indeed, none of the cited works (refs 42-46) are whole-brain modeling, rather these are constrained as coupled a few neural mass models (coupled cortical microcolumn or thalamocortical system but not at the whole-brain scale).

We have corrected that this sentence, stating instead these are neural mass models.

Line 208: what is the justification for using linear filters for the signal transmission? Nonlinear models have been shown to be crucial for modelling the brain at rest or

pathologies (Sanz Leon et al 2015) such as epilepsy (Hashemi et al Neuroimage 2020). A recent normalization of the connectome (Petkoski & Jirsa 2022 Netw Neurosci), showed that time-delays can be responsible for the appearance of PDR.

Our choice of linear filters was aimed at effectively capturing salient neurodevelopmental features across developmental time scales, while offering a parsimonious yet informative framework. We recognize that nonlinear models potentially provide more physiologic accuracy compared to their linear approximations, especially in capturing complex neural dynamics as highlighted by Sanz Leon et al. and the work of Petkoski & Jirsa. However, given that brain development proceeds on protracted temporal scales measuring months to years, the spatial and temporal precision of nonlinear methods may be less critical. While our linear approximation affords tractability and interpretability, we agree that future work could benefit from incorporating nonlinear dynamics may provide a more comprehensive understanding of neurodevelopmental changes in spectral brain activities.

In the case of EEG, using the linear models, it is time delay that make it feasible to get multiple frequency bands such as alpha and beta. How one can get simultaneous EEG peaks with linear models and no time delay?

While time-delay contributes to the appearance of the broadband spectrum, as the SGM transfer function has multiple complex conjugate pole pairs, each of which may correspond to a different resonant frequency.

Line 216 and 221; with a coupled linear delay differential equation, one can get both aperiodic and periodic components and this not a specific feature of the model used in this study and can be seen also in other neural mass modelling (even not at the whole brain level, rather thalamo-cortical system such as Hashemi et al Neuroinformatics 2018).

We appreciate the reviewer pointing out this previous work, which we have now cited. We have edited this section extensively and place emphasis on how our approach diverges from prior work in proposing a mechanistic framework for how EEG spectral maturation is directly linked to age-dependent tuning of this brain-wide network synchronization.

Line 307: Please see Lavanga, Mario, et al. "The virtual aging brain: a model-driven explanation for cognitive decline in older subjects." bioRxiv (2022).

We have added reference to Lavanga et al.'s recently published work.

Typo line 324, space after).

We have corrected this.

Line 326: I see it for 16000 simulations in appendix.

We have replaced this figure with Supplementary Figure 5, which provides additional comparison between SNPE-C and TSNPE.

Reviewer #2 (Remarks to the Author):

Summary:

A major goal of computational cognitive and neuroscience is to produce models with few parameters which can account for significant aspects of behavioral, neural or physiological data, making otherwise hard to discern patterns interpretable and hence understandable. Bernardo et. al. contribute to this general agenda a computational account of developmental aspects of neural activity patterns. The authors perform simulation-based inference on EEG spectral features with the Spectral Graph Model (SGM), and identify parameters of this model that consistently vary with age. They demonstrate that varying specific SGM parameters can recapitulate shifts in the Posterior Dominant Rhythm (PDR) and spectral slopes in accordance with maturation patterns observed in empirical studies, suggesting a mechanistic explanation for the spectral maturation of neural activity.

Main review:

Motivated by an analysis of the indeterminacy of the inverse problem (from data to parameters of the SGM) the authors opt for approximate bayesian inference to fit the SGM to subject data. We consider the findings as surprising, significant and an important contribution to the developmental part of the computational neuroscience literature. A computational account, which lets us, through the lens of a generative model and specifically interpretable parameter variations, appreciate important aspects of brain maturation can form significant connective tissue amongst a plethora of empirical findings across age groups.

In the following we however would like to point out some, mostly methodological, critiques which we would like to see addressed to bolster the credibility of the stated results. We address the major points below and then follow with a list of minor comments.

We deeply appreciate the feedback of the Reviewer. We have added SBC analyses as well as Regression diagnostic analyses which have improved the robustness of our findings. We address the Reviewer's concerns in a point-wise manner below.

SBI analysis:

1. In general the sections concerning SBI analysis could benefit from clearer motivation. What about the inference problem necessitates simulation based inference to begin with? The main SGM paper cited (Raj. et. al. 2020) does not emphasize simulation based inference.

We appreciate the reviewer's feedback here and have added emphasis on the motivation for utilizing SBI. We now discuss intractability of calculation of the likelihood function as a motivation for SBI in sections referencing SBI. Furthermore, we recently found other approaches including simulated annealing and gradient-based MCMC, however, are computationally inefficient compared to SBI. (Jin et al. Neuroimage 2023).

Raj et. al. (2020) moreover emphasize that the SGM is deterministic with a closed form solution in the Fourier domain. However, if the model is deterministic and uniform priors are used (which we understood the authors do use), a posterior should in fact be uniform over the region of admissible parameters. Posteriors illustrated in the paper do never seem uniform (and therefore should be rather bad approximations of true posteriors), however it is reasoned that the simulation budgets were sufficient. Can you clarify this?

The posteriors distribution obtained with SNPE derived from individualized spectra are non-uniform as they are consistent with both the prior and the observed data.

We have added Supplementary Figure 5, which provides further justification for the simulation budgets used for SNPE and TSNPE.

As also suggested below in the UMAP critique, parameter recovery studies could help the reader understand the quality of the posteriors. A useful technique to test your inference pipeline is 'simulation based calibration' (Talts et. al., 2018). Lastly, given the above, you may also resort to the MAP estimation technique of Raj. et. al. (2020), as a comparison.

We have added parameter recovery to our analysis. Following Boelts et al. (Elife 2022), we evaluated whether the inference procedure accurately recovers the ground-truth parameters (Supplementary Figure 4), and also demonstrate the effect of increasing simulation budget size in parameter recovery (posterior z-scores) in Supplementary Figure 5.

In addition, we have performed simulation-based calibration (SBC) to assess posterior calibration. We found that TSNPE provided improved calibration compared to SNPE-C (Supplementary Figure 7). TSNPE yielded uniformly distributed ranks with Kolmogorov-

Smirnov (KS) p values for τ_e : >0.5 , τ_i : 0.147, α : >0.5 , S : >0.5 , G_{EI} :0.0516, G_{II} : >0.5 , τ_G : 0.217; whereas KS test p values for SNPE-C fell below 0.05, indicative of non-uniform rank distribution. In addition, we performed the Classifier 2-Sample Test (C2ST), which demonstrated that the Data Average Posterior (DAP) compared to the prior demonstrated C2ST values between 0.473 to 0.510, approximating 0.5, indicating similarity of the DAP distribution to the prior distribution. We have added these results to the Supplementary Section and details on implementation to the Methodology.

2. As per Figure 3, the authors use the peaks of marginal posterior distributions as the estimates of parameters on which later regression models are based. The vector of peaks of posterior marginals may in fact not represent the MAP vector. This choice should be motivated more strongly. Moreover, information about the full posteriors is essentially never used (for example, why not run the regression analysis on the basis of posterior mean estimates?).

We have revised Figure 3 to display posterior mean estimates, which is more commonly used in the literature.

3. Figure S2 shows examples of posteriors under increasing simulation budgets. If the motivation of this figure is to show how posteriors stabilize after a simulation threshold is crossed, it fails to convey this message. In fact the posterior distribution for 8000 simulations looks quite different from the posterior distribution which used 16000 simulations. To illustrate its (assumed) point, the figure should show graphs based on simulation budgets which cross the point at which no further change in the posterior is elicited from increases in the simulation budget.

The effect of increasing simulation budgets with SNPE-C compared to TSNPE is now demonstrated in supplementary figure 5, which demonstrates stabilization and more ideal posterior contraction of the posteriors, particularly with TSNPE.

4. The authors motivate their choice of the data summary statistic by stating that the first periodic and aperiodic components of the SGM spectral output “succinctly parameterizes physiologically and developmentally pertinent EEG spectral features”. This is reasonable, but also heuristic. Given that all their analyses and conclusions rely heavily on this choice, it seems important to characterize how sensitive the inference procedure is with respect to this choice.

To address the issue of sensitivity in our inference procedure with respect to the choice of data summary statistic, in our revision, we now conduct the analysis using binned Power Spectral Density (PSD) as the summary statistic. This choice is motivated by previous work in EEG modeling, specifically Rodrigues et al., who successfully utilized

binned PSD for neural mass model (NMM) inversion with Hierarchical Neural Posterior Estimation (Rodrigues, Pedro, et al. HNPE: Leveraging Global Parameters for Neural Posterior Estimation. NeuRIPS 2021.)

Regression analysis:

1. The regression analysis reported in Figure 4, opens some questions regarding fulfillment of basic assumptions of the regression models. For Figure 4a), 4b) and 4c) one can argue that the basis of the increase/decrease in parameter values across ages, might not stem from an actual increase in mean, but rather an increase in variance of dependent variables across age. The estimates of these variances are in fact available since the chosen parameter values are derived from posterior distributions.

Reanalysis with truncated SNPE, as opposed to SNPE-C/APT, demonstrated overall less change in variance across age. We applied Breusch-Pagan test for heteroscedasticity (Supplementary Figures 11 and 12) and discuss where this is expected to have an effect (or no effect) on our findings in the Results section.

2. The parameter values used for this analysis have to be considered in light of the critique of the SBI analysis. The regression was based on the maxima of the approximate marginal posteriors. Instead, the parameter values corresponding to the actual MAP estimates could have been used. Another standard approach is to consider the posterior mean estimates. It would be essential to see whether the findings are robust to these variations here.

We appreciate this important distinction. We have now reported posterior mean estimates, which is more commonly used in the literature. Overall, this did not appear to affect our primary findings.

3. In Figure 5 (b,c) the authors illustrate how a polynomial regression on relevant fitted model parameters fares in predicting the PDR and aperiodic exponents of observed data. To establish expectations, it will be useful to include a prediction of age and/or PDR and aperiodic exponents directly from the summary statistics. How good / bad is the level of inference afforded via the SGM parameters in comparison?

We now demonstrate a comparison utilizing PDR and aperiodic exponents automatically extracted utilizing the FOOOF method (Donoghue T et al. Nature Neuroscience, 2020). SBI with SGM polynomial regression model demonstrates improved performance in prediction of age compared to polynomial regression that utilizes FOOOF-derived PDR and aperiodic exponents. The FOOOF regression model outperformed the SBI-SGM regression model in prediction of PDR.

4. A polynomial regression of model parameters to predict main features (PDR, aperiodic exponent) of the EEG spectra from SGM parameters was only marginally successful. Is this because of the choice of summary statistic on which the posterior inference was based? Again it is unclear what level of success should be expected.

Prior literature demonstrates an R^2 between 0.3 to 0.7 with best results achieved utilizing machine learning approaches, with a mix of time-domain and frequency domain features, often with feature sets containing upwards of 30 or more features (Engemann, Denis A., et al. "A reusable benchmark of brain-age prediction from M/EEG resting-state signals." *Neuroimage* 2022). In comparison, the SGM approach demonstrates comparable performance utilizing a biophysically-principled model with 7 parameters, thus providing improved model transparency and insight into underlying brain dynamics correlated with aging and development. We have added this comparison to the Results section.

UMAP analysis:

The UMAP analysis is used to argue that the SGM may exhibit indeterminacy issues. This argument seems circular. Instead, some parameter recovery studies (perhaps from synthetic data simulated using SGM) could be performed which illustrate e.g. strong parameter trade-offs if data is specified via summary statistics as chosen in the paper. These parameter studies can also inform some of the interpretations regarding roles of respective parameters implicit in the (forward simulation) analysis presented in Figure 2. Figure 2 can only show univariate claims (varying one, hold other parameters constant) which do not necessarily elucidate more complicated parameter tradeoffs that could emerge when dealing with the inverse problem (parameters might compensate in complex fashion to produce given patterns).

We now provide Supplementary Figure 5 with Bayesian sensitivity analyses which demonstrates ideal posterior contraction and identification across all parameters with the presented summary statistics. Additionally, we have introduced binned PSD as a summary statistic, following Rodrigues et al. (Rodrigues, Pedro, et al. "HNPE: Leveraging Global Parameters for Neural Posterior Estimation." *Advances in Neural Information Processing Systems* 34 (2021): 13432-13443).

Minor comments:

73-75, discussion of 2B refers to 2C and vice versa

We have corrected this.

81-82, emphasis should be on “variation in parameters” of the SGM that can account for developmental trajectories.

We have added changed this sentence to emphasize that SGM parameter variation may account for spectral developmental trajectory.

95-96, the neural density estimators (NDEs) are trained to provide approximate posteriors. Granted if you use uniform priors in principle NDEs should provide you with a uniform posterior over admissible parameters but only if your model is noise-free and suffers from any parameter non-identifiability (degeneracy). The wording here is misleading.

We appreciate the insightful comment and agree that our initial wording conveyed an oversimplified view of the relationship between priors and posteriors. We have clarified this sentence as follows: “To further elucidate key parameters driving EEG spectral development, we utilized a Bayesian approach employing the SBI neural density estimator to identify approximate posterior distributions of SGM parameters that best align with empirical EEG spectra.”

141-143 The accuracy of the posterior does not measure model performance, the posterior predictive may. If “model” here indicates the NDE (so the approximate posterior itself), this should be specific to avoid ambiguity.

We appreciate the feedback that the usage of “model” in this paragraph was unclear. We have revised to paragraph to place more emphasis on intractability of the likelihood function in motivating the SBI approach.

193 Two models being special cases of each other seems logically implausible.

We have clarified this and now state that the “series expansion and eigenmode approaches yield highly similar mappings between functional and structural networks.”

206-208 If linear models actually perform better than non-linear models, then no extra motivation is needed to use them.

This is an important point and we adopt the stance that there are advantages and disadvantages to both nonlinear and linear models. We have altered our messaging on this point and point out that the advantageous interpretability and computational tractability of linear models such as the SGM should be seen as a complement to, rather than a replacement for, nonlinear methods.

308-310 The fact that multiple parameter combinations may be able to generate similar data is not in and of itself what makes direct calculation of the posterior problematic. If

you do not have a likelihood function but can only simulate from the model, you cannot “calculate the posterior” as in you cannot evaluate it and need to resort to likelihood free methods (SBI for example). If you do have a likelihood you may want to use MCMC to sample from the posterior, which is complicated by multimodality and can be complicated by high correlations (two facets of indeterminacy), but you can evaluate the posterior or an object proportional to it.

We appreciate this feedback and have corrected this paragraph to clarify the motivation behind SBI is intractability of the likelihood function.

590 (Pg. 28) Axis titles are too small. Figure S1 a), age in years not days.

We have re-formatted Supplementary Figure one to correct these issues.

Reviewers' comments:

Reviewer #1 (Remarks to the Author):

I enjoyed reading the paper, and I acknowledge the authors for incorporating extensive analysis. Most of my questions have been addressed appropriately in the revision, but there are some points that could further enhance the paper, as follows:

1) It appears that the authors concluded that there are no changes in the connectome, attributing the observed alterations in EEG data to functional changes. However, in Bethlehem Nature 2022, substantial changes in anatomical data can be observed even within the ranges utilized in this paper. How can this explanation accommodate the structural-functional relationship? I leave this question to the authors, but it's worth noting that in the literature, there is evidence-based emphasis on the role of the connectome, particularly in explaining aging.

2) In Section 2.5, the conclusion is that determining the superiority of either method is inconclusive. This is totally fine as this stems from factors like data complexity, modeling assumptions, and inherent methodological trade-offs. Both methods, despite strengths, have limitations, making a straightforward comparison challenging.

I acknowledge the fact that you fitted empirical question using the both methods.

Figure S4's right column illustrates TSNPE with three rounds and 2000 simulations?

Because except Fig S4, it seems that the rest of results support the use of TSNPE.

3) From the synthetic data analysis, it appears that achieving a perfect recovery is challenging due to degeneracy, characterized by a high relationship between parameters. To enhance the understanding of this issue, could the authors provide an example featuring the observed and fitted synthetic EEG, alongside the joint posterior, similar to Figure 3? Such an illustration would be valuable in revealing potential correlations between parameters.

In the case of a high correlation between certain parameters, caution is warranted in interpreting changes in empirical data. Changes in one parameter might be linked to changes in others. For readers, understanding the correlation structure of model parameters is crucial. A plot of joint posteriors would effectively address this concern, providing valuable insights into the interdependence of parameters.

4) Section 2.6. Line 146: I am not sure to call the fitted data in Fig 3 as "closely" resemble the input empirical EEG.

5) In Section 4.4, while the majority of literature employing SBI often cites the intractability of likelihood as the primary motivation, it's noteworthy that the authors have nicely and successfully derived an analytical solution to the EEG.

As demonstrated by Hashemi et al. in NeuroImage 2020, Hamiltonian Monte Carlo (HMC) can be effectively applied even at the whole-brain level, providing superior scalability with the number of parameters compared to SBI, albeit at a much more computational costs.

My humble suggestion is to include a reference to the differentiability (e.g., Hashemi et al, NeuroImage 2020). In SBI, neither the model nor data features need to be differentiable. On the contrary, when utilizing automated HMC in advanced tools such as Stan, the challenge for such models lies in handling complex numbers that emerge in the analytical solution. Otherwise, HMC can easily handle the dimension of model parameters used in the article.

Reviewer #2 (Remarks to the Author):

General comment:

The authors have submitted a responsive revision, and the paper is more compelling with these revisions. Our major qualm with the initial presentation of the inference machinery (SBI) and the respective reporting of results has been alleviated by a much-improved presentation. Some minor criticisms remain (delineated below), which the authors can address in a subsequent revision.

Major comment:

One remaining concern that the authors can address pertains to the differential results from SNPE-C and TSNPE. From Figure 4, one can observe that SNPE-C and TSNPE have complementary weaknesses, depending on the parameter. It may be worthwhile to explore at least one other model family (such as “BayesFlow”) to compare against – especially since the paper's main takeaways rely on the goodness of parameter estimation. We don't insist that the authors specifically use BayesFlow or any other specific method, but they can either consider these or just discuss the complementary advantages and weaknesses and acknowledge that other methods might improve further. It would also be useful to comment on what one can infer from real data depending on the parameters the researcher are interested in, and perhaps to discuss more the root source of why one method recovers some parameters better than the other and vice versa.

Additionally, the authors can perhaps make the posterior variances (normalized per subplot) explicit (via a color scheme, for instance) and see if any obvious patterns emerge. Reporting the relative average posterior variance across parameters would also lend clarity.

While SNPE-C has trouble with the S parameter, TSNPE seems to have trouble with G_{EI} , G_{II} , τ_G , and τ_i . The parameters that TSNPE has trouble with seem to be the ones that are not well identified in the posterior pair plots from Figure 3. In both cases, the overall fits are well below the ceiling expected variance explained (from the numbers the authors quote in the rebuttal). Any discussion and numerical proof of the stability of the reported findings would be greatly beneficial.

Minor comments:

Figure 3: Do the red dots represent ground truth parameters of the SGM model? How did the authors have access to “ground truths” when dealing with EEG data from subjects? Additionally, we urge the authors to be careful when using terms of art. For instance, in the caption of Figure 3, “SGM Parameter Posterior Likelihoods,” is confusing since “Likelihood” is a term of art that conveys a specific meaning. It is perhaps better to say “Posterior distributions of SGM parameters” here.

120 - 121: “On running ...” → “Running”

122: “most accurate recovery error”: Is it perhaps better to state this as the “best R^2 ”?

128-141: It is helpful to avoid vague terminology such as “accurate parameter recovery.”

193: “We evaluated regression model predictive capacity” → We evaluated the predictive capacity of the regression model

431: “Bayesian joint distribution” → unclear what this refers to, maybe spend a sentence or two

434: “the data does not informatively alter” → the data does not inform [...]

439: The posterior z-score seems to depend on over/under dispersion of the posterior. If the posterior is overdispersed, then z-scores become lower and vice versa. This seems like a metric that could be more useful if you do MCMC on analytical likelihood functions. SNPE and other posterior amortizers can result in misleading results on this front.

Response summary:

We appreciate the insightful feedback provided by the Reviewers, which we have incorporated into our revision. In addition to the changes outlined in our response, we note the following minor changes to our manuscript:

1) We now refer to SNPE as NPE. We had originally adopted the SNPE naming convention used in Gonçalves et al. (Elife 2020), referring to SNPE in the context of single-round inference. Following Desitler et al. (NeuRIPS 2023), we now refer to SNPE as NPE, drawing distinction from TSNPE and reflecting its non-sequential aspect more accurately.

2) We have moved the bulk of Results Section 2.5 which comprised parameter recovery and simulation-based calibration results into Supporting Information, which now includes analyses and discussion of the performance differences seen with different SBI methods. In Results Section 2.5, we provide a summary of these analysis with references to Supporting Information.

We look forwards to addressing any further feedback that the Reviewers may have.

Reviewers' comments:

Reviewer #1 (Remarks to the Author):

I enjoyed reading the paper, and I acknowledge the authors for incorporating extensive analysis. Most of my questions have been addressed appropriately in the revision, but there are some points that could further enhance the paper, as follows:

1) It appears that the authors concluded that there are no changes in the connectome, attributing the observed alterations in EEG data to functional changes. However, in Bethlehem Nature 2022, substantial changes in anatomical data can be observed even within the ranges utilized in this paper. How can this explanation accommodate the structural-functional relationship? I leave this question to the authors, but it's worth noting that in the literature, there is evidence-based emphasis on the role of the connectome, particularly in explaining aging.

We appreciate the Reviewer's point that structural alterations arise early in the lifespan and remain essential throughout aging. Also, we acknowledge that our revision underemphasized the importance maturational changes in the structural connectome which are critical for normal development. In our revision, we more clearly delineate the two main effects during brain maturation: evolution of the structural connectome and of the structure:function relationship itself. While the former has been explored (Bethlehem et al., Nature 2022), the latter remains an open question that we attempt to address in our manuscript. We have revised sections of our paper as follows:

Abstract Discussion, lines 10-23, (we have removed emphasis that spectral maturation may be predominantly driven by the structural:functional relationship):

“These results suggest that spectral maturation of brain activity observed during normal development is supported by functional adaptations, specifically age-dependent tuning of localized neural dynamics and their long-range coupling within the macroscopic, structural network.”

Introduction (17-23), we have emphasized the importance of brain morphology changes over the lifespan (Bethlehem 2022):

“Morphological transformations of the brain across the lifespan are well established (Bethlehem 2022), and underpin well-described structural and functional network modifications occurring at various temporal and spatial scales during development (Zamani 2022). Yet, the mechanisms linking morphological and structure-function remodeling to electrophysiological developmental changes is unknown. This gap is critical, as deviations from the typical electrophysiological maturation are associated with a spectrum of developmental disorders, including autism and epilepsy (Bozzi 2018, Nelson 2015, Sohal 2019). Here, we apply simulation-based inference (SBI) (Goncalves 2019) in conjunction with the spectral graph model (Raj 2020) to investigate structure-function dynamics that guide developmental EEG maturation.”

Results, Figures: We have moved our analysis comparing effect of adult versus neonatal connectome on synthetic EEG spectra to Results Figure 3, to add emphasis on the importance of structural connectome in shaping the structure:function coupling.

Discussion (Conclusion, lines 350-355), we have emphasized the importance of changes in morphological and structural connectivity as follows:

“While the critical trajectories of morphological and structural connectivity remodeling during development are well-established, the structure-function relationship remains to be clarified. Our findings suggest that the evolving structure-function interplay influences canonical features of brain spectral maturation, including the evolution of PDR and 1/f aperiodic activity. This interplay is inextricably linked to changes in brain morphology and structural networks, yet understanding the precise mechanisms underlying this connection warrants additional investigation.”

2) In Section 2.5, the conclusion is that determining the superiority of either method is inconclusive. This is totally fine as this stems from factors like data complexity, modeling assumptions, and inherent methodological trade-offs. Both methods, despite strengths, have limitations, making a straightforward comparison challenging.

I acknowledge the fact that you fitted empirical question using the both methods.

Figure S4's right column illustrates TSNPE with three rounds and 2000 simulations?

Because except Fig S4, it seems that the rest of results support the use of

TSNPE.

Yes, Figure S4 (now Fig S6 in the revision) right column illustrates TSNPE with three rounds and 2000 simulations. We have clarified that the simulation budgets used in Figure S4 as follows: “The middle column shows parameter recovery with NPE utilizing 1000000 simulations and the right column shows parameter recovery with TSNPE utilizing three rounds and 2000 simulations.”

We have provided more discussion on the different performance seen between SBI methods and their potential causes in the Supporting Information Section. We also summarize these differences in the Results Section 2.5, Lines 113-130:

3) From the synthetic data analysis, it appears that achieving a perfect recovery is challenging due to degeneracy, characterized by a high relationship between parameters. To enhance the understanding of this issue, could the authors provide an example featuring the observed and fitted synthetic EEG, alongside the joint posterior, similar to Figure 3? Such an illustration would be valuable in revealing potential correlations between parameters.

In the case of a high correlation between certain parameters, caution is warranted in interpreting changes in empirical data. Changes in one parameter might be linked to changes in others. For readers, understanding the correlation structure of model parameters is crucial. A plot of joint posteriors would effectively address this concern, providing valuable insights into the interdependence of parameters.

We have provided plots similar to Figure 3 demonstrating observed and fitted synthetic EEG alongside the joint posteriors in Supplemental Figures 4-6 for NRE, NPE, and TSNPE. In addition, we evaluated Pearson correlation coefficient matrices (Supplementary Figure 13) to investigate the correlation structure of model parameters. We provide discussion on these findings in the Supporting Information Section 9 lines 896-907 as follows:

“Next, upon inspection of posterior predictive checks (PPC) utilizing synthetic spectral realizations we observed that the estimated posterior distributions qualitatively demonstrated interdependencies among certain SGM parameters which varied according to SBI method utilized (Supplementary Figures 3-5). We assessed these correlations by computing Pearson correlation coefficient matrix for the joint marginal distributions (Supplementary Figure 13). There were strong correlations across several joint marginals of the estimated posterior distribution, indicative of model degeneracy, wherein multiple model parameterizations yield similar spectral realizations. NPE had increased sensitivity to detect correlations between parameters compared to TSNPE. Increasing simulation budget size with TSNPE led to increased alignment in detected significant correlations with NPE, suggesting that increased simulation budget size for neural density estimator training increases its ability to capture interdependencies in the multivariate posterior distribution. When averaging across the entire synthetic dataset, no significant correlations were observed. This observation suggests that while model degeneracy may explain the correlations seen in individual joint marginal distributions, it

may not manifest uniformly against the variance representative of spectral developmental trajectories.”

4) Section 2.6. Line 146: I am not sure to call the fitted data in Fig 3 as "closely" resemble the input empirical EEG.

We acknowledge this feedback and removed “closely” for accuracy.

5) In Section 4.4, while the majority of literature employing SBI often cites the intractability of likelihood as the primary motivation, it's noteworthy that the authors have nicely and successfully derived an analytical solution to the EEG. As demonstrated by Hashemi et al. in NeuroImage 2020, Hamiltonian Monte Carlo (HMC) can be effectively applied even at the whole-brain level, providing superior scalability with the number of parameters compared to SBI, albeit at a much more computational costs.

My humble suggestion is to include a reference to the differentiability (e.g., Hashemi et al, NeuroImage 2020). In SBI, neither the model nor data features need to be differentiable. On the contrary, when utilizing automated HMC in advanced tools such as Stan, the challenge for such models lies in handling complex numbers that emerge in the analytical solution. Otherwise, HMC can easily handle the dimension of model parameters used in the article.

We appreciate this point, highlighting recent advances in differentiable simulation models utilizing automated differentiation tools such as HMC. We have added the following to our Methodology, Section 4.4 lines 443-453:

“While we focus on recent SBI methods here, we acknowledge there are alternative robust methods for parameter estimation. A prominent Bayesian inference approach is the Hamiltonian Monte Carlo (HMC) algorithm, which offers superior convergence and robustness in handling high-dimensional spaces and models with highly correlated parameters relative to traditional MCMC methods (Duane et al., Betancourt 2014b). Whereas in SBI, neither the model nor summary statistics features are required to be differentiable, HMC necessitates model differentiability, as it employs gradient calculations of the log posterior with respect to the model's parameters to compute the Hamiltonian dynamics, facilitating efficient exploration of the posterior distribution. A compelling demonstration of these gradient-based HMC methods in combination was demonstrated by Hashemi et al. for the inversion of a differentiable whole-brain model to model subject-specific epileptogenicity. (Hashemi et al. Neuroimage 2020) The SGM is differentiable; however, caution and further investigation are needed to evaluate the stability of eigendecomposition gradients due to parameter degeneracy and the use of non-Hermitian matrices in the SGM.”

Reviewer #2 (Remarks to the Author):

General comment:

The authors have submitted a responsive revision, and the paper is more compelling with these revisions. Our major qualm with the initial presentation of the inference machinery (SBI) and the respective reporting of results has been alleviated by a much-improved presentation. Some minor criticisms remain (delineated below), which the authors can address in a subsequent revision.

Major comment:

One remaining concern that the authors can address pertains to the differential results from NPE and TSNPE. From Figure 4, one can observe that NPE and TSNPE have complementary weaknesses, depending on the parameter. It may be worthwhile to explore at least one other model family (such as “BayesFlow”) to compare against – especially since the paper's main takeaways rely on the goodness of parameter estimation. We don't insist that the authors specifically use BayesFlow or any other specific method, but they can either consider these or just discuss the complementary advantages and weaknesses and acknowledge that other methods might improve further.

We acknowledge the importance of evaluating differential performance between NPE and TSNPE, and the benefit of including an additional model family. We performed additional analyses, including neural ratio estimation (NRE), as stated below:

Methods section 4.4, lines 434-439:

“In our approach, we evaluated three recent SBI techniques: Neural Ratio Estimation (NRE), Neural Posterior Estimation (NPE), and Truncated Sequential Neural Posterior Estimation (TSNPE). In NRE, a neural network classifier is trained to approximate the likelihood-to-marginal ratio (Hermans et al. 2020)...”

Supporting Information Section 9.1, lines 851-868: “...we more accurately recovered SGM parameters with the NPE and TSNPE, relative to NRE (Supplementary Figure 3-6). Alpha was the most accurately recovered parameter across all model families, whereas conduction speed and tau_G had the poorest recovery. For parameter recovery across all SGM parameters, NPE had the lowest mean relative estimated error (REE), 0.0451, compared to 0.0560 for TSNPE and 0.114 for NRE (Supplementary Figure 6). To evaluate potential pathologies of the SBI process, we performed Bayesian sensitivity analysis evaluating posterior Z-score and posterior contraction (Supplementary Figure 7) for each parameter using NRE, NPE, and TSNPE across various simulation budget sizes (Betancourt 2018). We demonstrate that relative to NPE and NRE, TSNPE provided improved posterior contraction (Supplementary Figures 8 and 9). Systematic posterior underdispersion or overdispersion limited usefulness of posterior z-scores in the comparison of different model families and their respective parameterizations, particularly for NRE, whose posteriors demonstrated significantly increased posterior dispersion relative to NPE and TSNPE. We evaluated posterior dispersion indices (PDI) quantifying the degree of posterior dispersion while standardizing for the varied scale of different SGM parameters found that NPE and TSNPE had reduced PDI relative to NRE (Supplementary Figure 10).

To evaluate the calibration of the uncertainties of the estimated posteriors, we assessed the SBC of NPE and TSNPE. NRE was excluded from SBC analysis because the NRE-based inference of SGM parameters proved computationally prohibitive due to the utilization of MCMC sampling. We found that the TSNPE yielded posteriors with well-calibrated uncertainties, whereas NPE yielded posteriors with left-skewed rank distribution consistent with systematic underestimation of the posterior means (Supplementary Figure 11)..."

It would also be useful to comment on what one can infer from real data depending on the parameters the researcher are interested in...

In terms of what researchers may be able to infer, we have added the following to the Discussion section 3.8, Implications and Future Directions lines 338-349:

"The ability of SBI capture age-dependent changes in SGM parameters from empirical EEG data suggests broader applicability. For instance, in epileptogenesis, elucidating the time-dependent slow dynamics of long-range coupling and excitatory:inhibitory balance may provide insights into pathologic neurophysiological shifts that engender the development of epilepsy, whereas identifying rapid changes in these parameters could elucidate dynamics of pre-ictal and ictal states. Similarly, deviation of these parameters from typical trajectories could be used to investigate spectral differences that arise in autism and other neurodevelopmental disorders. Lastly, recent work by Lavanga et al. has utilized SBI in conjunction with whole brain modeling to understand structure-function relationship underlying cognitive decline in aging(Lavanga et al. 2023) Similarly, the SGM, which has been recently used to model abnormal neural oscillations and their cellular correlates in patients with Alzheimer's Disease (Ranasinge et al. 2022), in conjunction with SBI offers an analytical framework to further investigate mechanisms underlying cognitive decline..."

...and perhaps to discuss more the root source of why one method recovers some parameters better than the other and vice versa.

We have performed additional analyses to evaluate sources of the differential performance of NPE and TSNPE. We summarize performance differences in the SBI methods in Results Section 2.5, Lines 121-129:

"NRE, NPE, and TSNPE demonstrated differential performance across parameter recovery and SBC, with NPE and TSNPE generally outperforming NRE (Supplementary Figures 3-10). While NPE demonstrates robust parameter recovery, it tends to produce poorly calibrated posterior distributions (Supplementary Figure 11) and is prone to posterior leakage at the parameter bounds (Supplementary Figure 12). In contrast, TSNPE has well-calibrated posteriors relative to NPE; however, its scalability is limited under conditions of extensive simulation requirements and large datasets. We discuss potential sources of the differential performance across SBI methods further in Supporting Information 2. Given the general preference for conservative over overconfident posterior estimates—the latter potentially leading to erroneous scientific conclusions (Ward 2022)—we utilize TSNPE in subsequent application of SBI-SGM to empirical data given its improved calibration results compared to NPE."

We evaluate reasons for different performances further in Supporting Information Section 9.2, lines 880-913, discussing posterior leakage seen with NPE and different ability of each SBI method to handle model degeneracy. In addition, we discuss the different simulation budgets used as a reason for improved parameter recovery with NPE compared to TSNPE in Supporting Information Section 9.2, Lines 908-913.

Additionally, the authors can perhaps make the posterior variances (normalized per subplot) explicit (via a color scheme, for instance) and see if any obvious patterns emerge. Reporting the relative average posterior variance across parameters would also lend clarity.

We have demonstrated posterior variance as measured by posterior dispersion indices across parameters in Supplementary Figure 10. In the S10 legend we state the following: “Posterior dispersion indices (PDI) were obtained by normalizing variance by respective parameter mean in order to account for variation in scales across SGM parameters. NPE and TSNPE had reduced PDI relative to NRE. NPE and TSNPE had similar PDI profiles. PDI across synthetic and observed datasets revealed highest PDI in conduction speed, excitatory:inhibitory gains; aligning posterior predictive check findings that these values had relatively more degeneracy.”

We also now state the following in the Supporting Information section, 9, lines 860-863: “We evaluated posterior dispersion indices (PDI) quantifying the degree of posterior dispersion while standardizing for the varied scale of different SGM parameters and demonstrate that NPE and TSNPE had reduced PDI relative to NRE (Supplementary Figure 10).

While NPE has trouble with the S parameter, TSNPE seems to have trouble with G_EI, G_II, tau_G, and tau_i. The parameters that TSNPE has trouble with seem to be the ones that are not well identified in the posterior pair plots from Figure 3. In both cases, the overall fits are well below the ceiling expected variance explained (from the numbers the authors quote in the rebuttal). Any discussion and numerical proof of the stability of the reported findings would be greatly beneficial.

We agree that discussion on the stability of the findings is beneficial and have included the following into our Discussion, Section 3.5, Lines 281-304:

“The stability or robustness of our findings hinge on several aspects: data characteristics and degree of model misspecification; SBI components such as prior specification and posterior approximator hyperparameterization (Bürkner et al. 2023); and the stability of the SGM itself. Regarding dataset characteristics, the experimental data distribution, noise levels, and alignment (or misalignment) of the model with the data-generating process may affect SBI robustness (Huang et al. 2024). EEG data demonstrates substantial variability across the developmental timeline. Thus, depending on the age group of interest, a more targeted analysis within narrower developmental windows may yield more robust inferences due to decreased variability and the option to use age-specific priors. Concerning stability considerations related to the SBI, our evaluations utilizing synthetic reference data demonstrated that stability and accuracy of

posterior distributions, given a priori knowledge of the true solution, depended on NPE hyperparameterization, such as the simulation budget utilized. In addition, one may consider approaches to improve SBI stability and reliability, including ensembling and post-hoc calibration methods (Hermans et al. 2021). Further work involving generalization studies on larger datasets is required to effectively evaluate the stability of the SBI-SGM framework.

Regarding numerical stability of the SGM model itself, Verma et al. applied root locus analysis to delineate the bounds of SGM parameters that give rise to dynamical behaviors, including damped oscillations, limit cycles, or unstable oscillations (Verma et al. 2023). Our SBI findings for excitatory and inhibitory time constants and alpha were within ranges that promote stable oscillatory activity. In contrast, excitatory:inhibitory gains were consequently inferred to be above boundaries that ensure stable oscillations, suggesting a potential for instability. These parameters exhibited increased posterior dispersion and variance in identifiability, which could reflect the unstable regime in SGM parameter space leading to greater unpredictability in SGM output. However, our empirical data encompassed subjects wherein oscillatory activity demonstrated fluctuations and instability over the monitoring duration, was intermixed within prominent 1/f activity, or was absent in the case of younger infants. The ability of the SGM to account for such diverse neural dynamics promotes robustness in the SBI-SGM framework, particularly in accommodating the unpredictability and instabilities of whole-brain network activity.”

We also acknowledge that our R² performance is lower compared to prior work by Engemann et al. 2022. We did not previously emphasize that our age distribution is significantly lower thus limiting direct comparison. We clarify our comparison to prior work to Engemann et al in the Results Section 2.8, Lines 194-199:

“Prior literature in predicting age from EEG has reported R² values ranging up to 0.61 for datasets including pediatric subjects with best results achieved utilizing machine learning approaches (Engemann et al.). Engemann et al. reported deep learning and feature-engineering models had R² values of 0.61 and 0.33, respectively, on the normal subgroup of the Temple University Hospital Abnormal (TUAB) EEG dataset (N=1385, mean age: 48.6 yrs with standard deviation 17.9 years). Given that TUAB predominantly consists of adult subjects with 43 subjects between 10-20 yrs and only three subjects in the 0-10 yrs, direct comparison with our findings, which focus on the developmental age demographic, is limited.”

Here, we note that we do not include the two other EEG-specific analyses in Engemann et al. for comparison as they do not include pediatric subjects; these included:

1. Leipzig Mind-Brain-Body dataset with 153 subjects divided into young and old subgroups: young subgroup ages 20-35 yrs, median 24 yrs and standard deviation (SD) 3.1 yrs; old subgroup (ages 59-74 yrs, median 67 yrs, SD 4.7 yrs). Feature-engineering and DL models on this dataset had R² of 0.51 and 0.67, respectively.

2. Cuban Human Brain Mapping Project with 282 subjects, mean 31.9 and SD 9.30: Feature-engineering and DL models on this dataset had R^2 of 0.11 and 0.18, respectively.

Minor comments:

Figure 3: Do the red dots represent ground truth parameters of the SGM model? How did the authors have access to “ground truths” when dealing with EEG data from subjects?

We clarify that the red dots represent mean posterior values of the posterior distribution. These were used to generate the corresponding simulated spectra shown in the left column.

Additionally, we urge the authors to be careful when using terms of art. For instance, in the caption of Figure 3, “SGM Parameter Posterior Likelihoods,” is confusing since “Likelihood” is a term of art that conveys a specific meaning. It is perhaps better to say “Posterior distributions of SGM parameters” here.

We have used more precise wording and have changed the caption to state: “Posterior Distributions”.

120 - 121: “On running ...” → “Running”

We have corrected this.

122: “most accurate recovery error”: Is it perhaps better to state this as the “best R^2 ”?

We have corrected this.

128-141: It is helpful to avoid vague terminology such as “accurate parameter recovery.”

We have improved the clarity of this phrase as follows (Supporting Information Section 9.1, lines 876-879): “The performance differences between NPE and TSNPE are characterized by superior calibration in TSNPE and lower relative estimation error in parameter recovery for NPE.”

193: “We evaluated regression model predictive capacity” → We evaluated the predictive capacity of the regression model

We have made this correction to improve the clarity of this phrase.

431: “Bayesian joint distribution” → unclear what this refers to, maybe spend a sentence or two

We have clarified this as stated in Methods Section 4.5 lines 471-474:

“Bayesian identifiability and sensitivity analyses, which assess the Bayesian joint distribution and inference process, are utilized to identify pathology in the inference approach and to characterize the performance of the inference procedure (Betancourt 2018). The Bayesian joint distribution is determined by the model configuration space and

the prior distribution: $\pi(y, \theta) = \pi(y|\theta)\pi(\theta)$ and is conditioned upon on an observation, \tilde{y} , to obtain the posterior distribution (Betancourt 2018).”

434: “the data does not informatively alter” → the data does not inform [...]

We have made this correction to improve the clarity of this phrase.

439: The posterior z-score seems to depend on over/under dispersion of the posterior. If the posterior is overdispersed, then z-scores become lower and vice versa. This seems like a metric that could be more useful if you do MCMC on analytical likelihood functions. NPE and other posterior amortizers can result in misleading results on this front.

We have raised the limitation of posterior z-scores in our study in the Supporting Information 9.1, Lines 858-860 as follows: “Systematic posterior underdispersion or overdispersion limited usefulness of posterior z-scores in the comparison of different model families and their respective parameterizations, particularly for NRE, whose posteriors demonstrated significantly increased posterior dispersion relative to NPE and TSNPE.”

REVIEWERS' COMMENTS:

Reviewer #1 (Remarks to the Author):

Thank you to authors for detailed responses. The revision presents satisfactory results and adequately addresses the questions.

Reviewer #2 (Remarks to the Author):

The authors have submitted a responsive revision and our main concerns are addressed. The presentation of SBI results and the overall discussion of the SBI methodology and pitfalls is significantly improved in the revised version. We appreciate the discussion of parameter identifiability issues in the supplementary materials.

We have a few minor comments that would be helpful to address in the final published manuscript:

- 443-454 The authors contrast HMC with SBI as an alternative approach to Bayesian Inference. This is not strictly wrong, but misleading. HMC is an algorithm to sample from any target distribution, however as presented here, at least to the best of our understanding, it obfuscates the crucial difference in methodologies, the fact that SBI doesn't expect availability of likelihoods, whereas HMC is usually applied in the standard setting of likelihood based posterior inference. NPE, SNPE etc. are methods to construct posteriors however final sampling can proceed via a variety of samplers (inc. HMC).

- 472-475 The authors use the term 'Bayesian joint distribution', which seems unusual. Moreover, suggested reformulation: 'Bayesian identifiability analyses, which assess'  'We apply Bayesian identifiability and sensitivity analysis to assess...'

Reviewer #1 (Remarks to the Author):

Thank you to authors for detailed responses. The revision presents satisfactory results and adequately addresses the questions.

We appreciate the Reviewer's helpful feedback and insights in the review process which have helped to significantly improve our work.

Reviewer #2 (Remarks to the Author):

The authors have submitted a responsive revision and our main concerns are addressed. The presentation of SBI results and the overall discussion of the SBI methodology and pitfalls is significantly improved in the revised version. We appreciate the discussion of parameter identifiability issues in the supplementary materials.

We appreciate the Reviewer's helpful insights on prior commentary, as well as their additional feedback below which further improves our work. We have addressed the concerns point by point as outlined below.

We have a few minor comments that would be helpful to address in the final published manuscript:

- 443-454 The authors contrast HMC with SBI as an alternative approach to Bayesian Inference. This is not strictly wrong, but misleading. HMC is an algorithm to sample from any target distribution, however as presented here, at least to the best of our understanding, it obfuscates the crucial difference in methodologies, the fact that SBI doesn't expect availability of likelihoods, whereas HMC is usually applied in the standard setting of likelihood based posterior inference. NPE, SNPE etc. are methods to construct posteriors however final sampling can proceed via a variety of samplers (inc. HMC).

To provide a more thorough contrast between HMC and SBI, while also acknowledging their potential for combined use, we have modified the Methods section with the changes highlighted below:

"In contrast to SBI, which enables approximate Bayesian inference even when the likelihood function is unavailable, HMC requires availability of the likelihood function, as it operates within a likelihood-based framework of Bayesian inference. Furthermore, model differentiability is optional with SBI whereas HMC requires it. HMC employs gradient calculations of the log posterior with respect to the model's parameters to compute the Hamiltonian dynamics, facilitating efficient exploration of the posterior distribution. Additionally, HMC may be utilized to sample the posterior constructed with differentiable, likelihood-based SBI techniques such as NLE. This approach facilitates robust sampling and efficiently scales to high-dimensional parameter spaces [Brandes et al 2024]."

- 472-475 The authors use the term 'Bayesian joint distribution', which seems unusual. Moreover, suggested reformulation: 'Bayesian identifiability analyses, which assess'  'We apply Bayesian identifiability and sensitivity analysis to assess...'

Following the Reviewer's suggestion, we have changed the original sentence (lines 472-475) to:

"We apply Bayesian identifiability and sensitivity analysis to assess pathology in the inference approach and to characterize the performance of the inference procedure (Betancourt et al. 2018)."

We have also removed the phrase containing "Bayesian Joint Distribution".